

# HighResClimNevada: a high-resolution climatological dataset for a high-altitude region in Southern Spain (Sierra Nevada)

Matilde García-Valdecasas Ojeda[1,2], Feliciano Solano-Farias[1], David Donaire-Montaño[1], Emilio Romero-Jiménez[1], Juan José Rosa-Cánovas[1,2], Yolanda Castro-Díez[1,2], Sonia R. Gámiz-Fortis[1,2], and María Jesús Esteban-Parra[1,2]

[1]Department of Applied Physics, University of Granada, Granada, 18071, Spain
[2]Atmosoheric Physics, Andalusian Inter-University Institute for Earth System Research, Granada, 18071, Spain

*Correspondence to*: Matilde García-Valdecasas Ojeda (mgvaldecasas@ugr.es)

**Abstract.** High spatiotemporal climate datasets are essential to assess the impacts of climate change in high mountains, where the climate is highly variable. However, these regions are characterized by a lack of climatic information, and if any, it is usually short, sparse, or incomplete. This work presents a new series of very high-resolution (1 km) gridded climate datasets for Sierra Nevada (SN), a mountain range classified as a double climate-change hotspot, as it is a semi-arid mountain range in the Mediterranean area that is particularly vulnerable to climate change. The database, called HighResClimNevada, consists of a set of climate data derived from a climate simulation using the Weather Research and Forecasting (WRF) model for the period from 1991 to 2022 and forced with the ERA5 reanalysis. HighResClimNevada provides hourly/daily primary climate variables (i.e., near-surface temperature, precipitation, near-surface relative humidity, surface pressure, surface net radiation, and wind speed), but also bioclimatic variables, extremes indices from the Expert Team of Climate Change Detection and Indices (ETCCDI), and precipitation-hour indicators, which were postprocessed using aggregated temperature and precipitation values from primary climate variables. To evaluate the database performance, HighResClimNevada temperature and precipitation values were compared with reference datasets from different sources. In general, HighResClimNevada captures reasonably well the spatiotemporal variability of raw temperature but also bioclimatic variables and extreme indices in SN. It displays comparable behavior to other climatic products but with a greater level of detail due to its higher spatial resolution. For precipitation, variable, more uncertain and difficult to characterize, HighResClimNevada exhibits a higher amount of precipitation when compared to station-based, coarse satellite-based, and reanalysis-based products. However, these latter present problems in characterizing precipitation in high mountain regions due to the scarcity of data in areas with high spatiotemporal variability, such as SN. The precipitation from HighResClimNevada is comparable to other climatic products like CHIRPS or CERRA-Land, which captures better the spatiotemporal variability in this region. These findings, therefore, suggest HighResClimNevada as a valuable long-term climate tool for a variety of applications, including land management, hydrometeorological research, flora and fauna phenology, and risk assessment. The reported datasets are freely available for download via Zenodo platform (García-Valdecasas Ojeda et al., 2024, https://doi.org/10.5281/zenodo.14052394) .



## 1. Introduction

Mountain areas are particularly vulnerable to climate change due to several causes. On the one hand, these regions are characterized by extreme climatic conditions, forcing species of flora and animals to adapt to this environment. Consequently, mountains are rich in biodiversity, including endemic species. However, this fact makes these regions vulnerable, as species that are high-altitude specialists may be less resilient under a changing climate. As a result, if the warming rate continues, natural ecosystems may suffer catastrophic consequences due to a decrease in biodiversity (La Sorte and Jetz, 2010) or a reduction of the habitats for many species (Parmesan, 2006).For example, the rising temperatures caused by climate change are leading to an upward migration of flora and fauna species (Parmesan, 2006). Subsequently, mountain biodiversity, particularly endemic species and those with limited dispersal capacity (Viterbi et al., 2013), is being affected by the emergence of invasive species. On the other hand, high altitude areas play a key role in providing water resources to ecosystems and humans (Viviroli et al., 2020). As a result of increasing economic and demographic expansion, these regions are suffering unprecedented water stress (Beniston, 2003).This is particularly true over Mediterranean regions, where mountains serve as the main source of fresh water for downstream communities (the so-called water tower), who rely heavily on it during the late spring and summer (Polo et al., 2020).

Moreover, mountains are undergoing rapid environmental change (Beniston et al., 2018; Gobiet et al., 2014), caused, among other factors, by the substantial increasing temperature trend that occurred in these areas. Many studies have reported elevation-dependent warming (EDW) due to different climate mechanism, such as the snow-albedo feedback (Pepin et al., 2019, 2022; Rangwala and Miller, 2012). Specifically, an increasing temperature trend (around 0.13 ºC per decade) has been found over European mountains such as the Alps since the 19[th] century (Begert and Frei, 2018), which has become more pronounced, especially after the 1980s, with a warming rate of 0.5 ºC per decade (Nigrelli and Chiarle, 2023). This increased trend has also been found in southern Europe, such as the Pyrenees (0.17 ºC per decade) and the Sierra Nevada (0.13 ºC per decade), where increasing patterns are more generalized for the minima (Esteban-Parra et al., 2022; Sigro et al., 2024). In the latter area, moreover, the drying trend in annual and winter precipitations (Esteban-Parra et al., 2022) is becoming a threat to the ecosystems that inhabit this region.

To understand the causes of many fundamental questions and applications in environmental research and ecology in a changing climate over high mountains, long and temporally consistent climate data with high resolution in space and time is critical (Hartmann et al., 2000). This is even more relevant in high mountains, where limitations in measurement equipment or sensor maintenance make obtaining adequate climate records extremely challenging. As a result, climate information in these regions is usually short, sparse, or incomplete, and therefore, how the climate is changing in these regions remains uncertain. In this regard, regional climate models (RCMs) have proven to be useful tools as they can provide climate information regularly in space and time, allowing the development of consistent long-term climate data in regions with difficult access. This is particularly true when working with an RCM with a high resolution ($\Delta x \le 4$ km). At these spatial resolutions, deep convection can be explicitly resolved by the RCM instead of being parameterized, a reason why RCMs at km-scale resolution are known

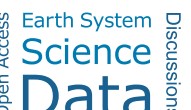

as convection-permitting models (CPMs) or convection-permitting RCMs (CPRCMs). This fact implies a significantly enhanced representation of orography and land-surface information. As a result, CPRCMs can better capture orographic precipitation and extreme rainfall events, as well as soil-atmosphere and cloud-radiation feedbacks and mountain snowpack (Coppola et al., 2020; Halladay et al., 2024; Lucas-Picher et al., 2024; Prein et al., 2015, 2017, 2020; Sangelantoni et al., 2022, among others), leading to a more realistic climate information.

This study describes a new series of 32-year high-resolution climate datasets for Sierra Nevada (SN), a Mediterranean mountain range in the Southern Iberian Peninsula (IP) that constitutes a double hotspot region. These datasets, called HighResClimNevada, provide primary climatic variables, bioclimatic variables, and extremes indices in a region with scarce climatic information with quality due to the difficulty of access. Furthermore, HighResClimNevada provides information at unprecedented spatial resolution in this region which is useful for studies focusing on the impacts of climate change on botany,

ecology, and other disciplines. HighResClimNevada, derived from climate modeling using a CPM, is available for the period from January 1991 to December 2022. This paper is structured as follows: Section 2 describes the model configuration for the development of HighResClimNevada, data used as a reference for the evaluation of HighResClimNevada, and variables provided by HighResClimNevada. Section 3 displays and discusses the results of the HighResClimNevada evaluation; Section 4 describes the data and code availability; and Section 5 summarizes and concludes the main findings of this work.



## 2. Methods

### 2.1. Study region

SN, located in the Baetic System (latitudes 36.93ºN to 37.20ºN and longitudes 3.53ºW to 2.65ºW), is the southernmost mountain range on the European continent. It houses some of the highest peaks in the IP, including Mulhacén (3479 m) and Veleta (3396 m)(Oliva et al., 2022). SN is a clear example of semi-arid high mountains being considered a double hotspot region where Mediterranean and Alpine climates cohabit about 40 km apart (Polo et al., 2019). Due to its singular conditions, it is home to many endemic plant and animal species, being considered one of the most important European hotspots for biodiversity (Blanca et al., 1998), and being a good candidate as a global change observatory. For these reasons, it was designated a Biosphere Reserve by UNESCO in 1986, Natural Park in 1989, and National Park in 1999.

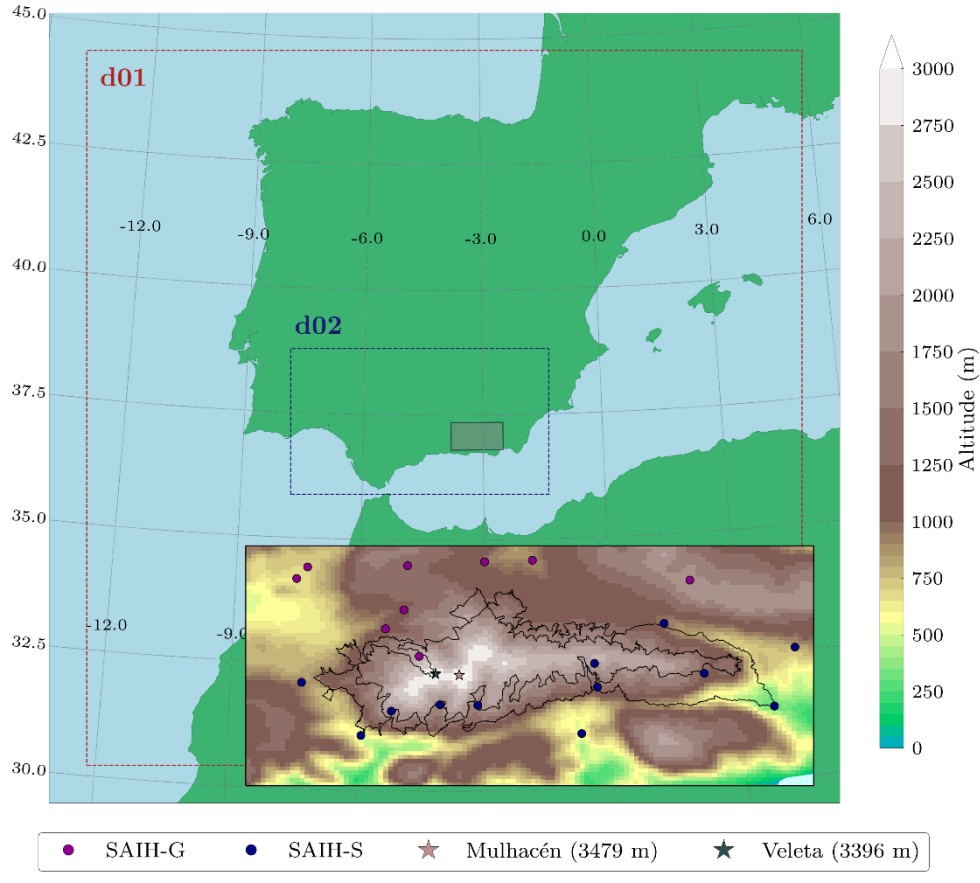

**Figure 1: WRF model configuration with two one-way domains: the parent domain (d01), which covers the Iberian Peninsula (IP) with 5 km spatial resolution, and the inner domain (d02), which spans the Andalusia region with 1 km spatial resolution. The area covered by HighResClimNevada is depicted in the inner figure at the bottom right, including the boundaries of both the Natural and National Parks of Sierra Nevada (black contours). The shading colors in this subfigure denote the elevation stated in meters above mean sea level, while the dots represent the locations of stations from the HidroSur automatic Hydrological Information System (SAIH-S) and the Guadalquivir automatic Hydrological Information System (SAIH-G).**





## 2.2.    Climate model

HighResClimNevada has been developed using the Weather Research and Forecasting Model (WRF) version 4.3.3 (Skamarock et al., 2021) in convection-permitting mode. To do that, a one-way double-nested configuration with a 5:1 nesting ratio (Fig. 1) was chosen: a parent domain (d01), which spans the whole IP with 5 km spatial resolution, and a nested domain covering SN with 1 km spatial resolution (d02). The Lambert projection was used with the center at 37.65ºN, 3.97ºW, and with 320 x 320 and 576 x 326 grid points in the west-east and south-north directions for d01 and d02, respectively. In the vertical, both domains were configured using 46 hybrid levels, with the top set to 50 hPa.

WRF was fed every 6 hours with the fifth generation of European ReAnalysis (ERA5, Hersbach et al., 2020), the European Centre for Medium-Range Weather Forecast's most recent reanalysis product, which has proven to be suitable for regional climate downscaling. To complete the entire 32-year period (1991-2022), the simulation was divided into 11-year runs, with the first year of each simulation being the spin-up of the model. This spin-up period was selected to balance the impact on simulations when a short spin-up period is used and the computational cost of CPM simulations. In this regard, a study conducted by Jerez et al. (2020) found that for atmospheric variables such as precipitation and temperature, a relatively short spin-up time (~ 1 week) is required. However, soil variables need longer periods to reach such an equilibrium, and it depends on the initial soil moisture initial conditions and soil depth, among others (Khodayar et al., 2015). In this context and considering the trade-off between suitability and computational resources, as well as the fact that this study uses ERA5 data as LBCs, one year as spin-up was finally used.

Concerning the selection of the physic schemes, we selected those indicated in the study conducted by Solano-Farias et al. (2024), where the model sensitivity was evaluated. From that work, the authors concluded that the combination of parameterizations composed by the WRF single-model 7-class (WSM7, Bae et al., 2019), Grell-Freitas (Grell and Freitas, 2014) for convection in the parent domain (d01), the Community Atmosphere Model 3.0 (CAM3.0, Collins et al., 2004) for both long- and short-wave radiation, the Asymmetric Convective Model version 2 (ACM2, Pleim, 2007) for planetary boundary layer (PBL), and the multiparametric Noah (NOAH-MP, Yang et al., 2011) land surface model (LSM) leads to an adequate characterization of climate conditions over Andalusia. A summary of the model configuration can be seen in Table 1.



**Table 1: Main characteristics of the WRF configuration applied to create HighResClimNevada.**

|  | Parent domain (d01) | Nested domain (d02) |
|---|---|---|
| **Regional climate model** | | |
| WRF-ARW v4.3.3 (Skamarock et al., 2021) | | |
| **Lateral Boundary Conditions** | | |
| ERA5 Reanalysis (Hersbach et al., 2023) every 6-hours | | |
| **Domain configuration** | | |
| *Nesting strategy:* | One-way | |
| *Spatial resolution:* | 5 km | 1 km |
| *Coverage:* | Iberian Peninsula | Andalusia |
| *Dimensions (grid points):* | 320 x 320 | 576 x 326 |
| *Vertical coordinate system:* | Hybrid | Hybrid |
| *Number of vertical levels:* | 46 | 46 |
| *Model top pressure:* | 50 hPa | 50 hPa |
| *Time step:* | 30 seconds | 6 seconds |
| **Parameterization schemes** | | |
| *Land surface model:* | Noah MP (Yang et al., 2011) | Noah MP (Yang et al., 2011) |
| *Planetary boundary layer:* | ACM2 (Pleim et al., 2007) | ACM2 (Pleim et al., 2007) |
| *Convection:* | GF (Grell and Freitas, 2014) | OFF |
| *Microphysics:* | WSM7 (Bae et al., 2019) | WSM7 (Bae et al., 2019) |
| *Long-wave radiation:* | CAM3 (Collins et al., 2004) | CAM3 (Collins et al., 2004) |
| *Short-wave radiation:* | CAM3 (Collins et al., 2004) | CAM3 (Collins et al., 2004) |
| *Atmospheric surface layer:* | Revised MM5 (Jiménez et al., 2012) | Revised MM5 (Jiménez et al., 2012) |
| *Spin-up time:* | 1-year | 1-year |

## 2.3. Reference datasets

Observational datasets, although valuable, are not error-free, particularly in mountainous regions, where uncertainties tend to be more pronounced (Prein and Gobiet, 2017). Thus, daily values of precipitation (pr) and maximum, mean, and minimum temperature (tasmax, tasmean, and tasmin, respectively) were used from different sources (Table 2) to evaluate the HighResClimNevada performance, avoiding drawing incorrect conclusions due to observational uncertainties.

Satellite-based precipitation estimations were considered as a source of precipitation information. The United States Global Precipitation Measurement (GPM) Team and the National Aeronautics and Space Administration (NASA) of United States developed the Integrated Multi-SatellitE Retrievals (IMERG) for GPM version 6 (Huffman et al., 2019, 2020). This half-hourly multi-satellite-based product provides precipitation estimates for almost the entire planet with a spatial resolution of 0.1º and for a period from 2000 to 2021. Among the different versions, IMERG Final run was selected, which is calibrated at a monthly scale using the 1º x 1º Global Precipitation Climatology Centre (GPCC) monthly monitoring product. On the other hand, the Satellite Precipitation Climate Prediction Center morphing method (CMORPH, Joyce et al., 2004) is a high-quality, high-resolution (~ 8 km of spatial resolution and a half-hourly time frequency) dataset developed by the National Oceanic and



Atmospheric Administration (NOAA) of the United States. CMORPH, which is derived from low orbiter satellite microwave measurements, is available for the period 1998 to near present for the latitudinal bands from 60 ºN to 60 ºS. The Climate

Hazards group InfraRed Precipitation with Stations data (CHIRPS, Funk et al., 2015) includes gridded products that provide precipitation estimates for latitudinal bands between 50ºS and 50ºN. This data combines satellite precipitation data from NASA and NOAA with in-situ station data from public and commercial archives. CHIRPS has been available since 1981, and we used the global daily CHIRPS v2 with 0.05º spatial resolution.

Additionally, gridded station-based products were used. On the one hand, meteorological gridded products developed by Peral

García et al. (2017) version 2 (referred to as ROCIO_IBEB) were used. These daily datasets are the result of interpolating high-quality observational time series from the Spanish Meteorological Agency's (AEMET) National Bank of Climatological Data. ROCIO_IBEB provides precipitation and extreme temperature (maximum and minimum) values for Peninsular Spain and Balearic Islands at a 5 km spatial resolution from 1951 to 2022. On the other hand, a monthly precipitation dataset (UGR-SNGrid) developed by the Atmospheric Physics Group at the University of Granada (Romero-Jiménez et al., 2024) was also

employed. UGR-SNGrid is a gridded dataset covering the SN Natural Park and its surroundings from 1990 to 2020, with a spatial resolution of two hundred m. For its development, precipitation records from the ClimaNevada database (https://climanevada.obsnev.es/) and the automated Hydrological Information System (SAIH) network were subjected to rigorous quality control before being spatially interpolated using the R package RegRAIN version 0.1.0 (Alzate Velásquez et al., 2017). More details in Romero-Jiménez et al. (2023).

Reanalysis data were employed as an additional source of information that would complement the sparse station network in this area. On the one hand, the global ERA5-Land reanalysis (Muñoz-Sabater et al., 2021) provides climatic and land-related variables with a 9 km spatial and hourly timestep. It was developed using ERA5 as atmospheric forcing and is available from January 1950 to the near present. Additionally, the Copernicus European Regional ReAnalysis products (CERRA and CERRA-Land) were used. Both regional products are high-resolution reanalysis developed using dynamical downscaling methods. The

CERRA reanalysis (Schimanke et al., 2021) employs ARMONIE-ALADIN as limited-area numerical weather model, which is driven by ERA5, and the CERRA system to assimilate a dense network of in situ observations and satellite information. CERRA-Land (Verrelle et al., 2022), on the other hand, employs the land surface model SURFEX v8.1, which is run offline forced using 3-hourly CERRA atmospheric variables and the daily surface precipitation from the MESCAN system. CERRA and CERRA-Land have the same integration domain (e.g., orography, coverage, etc.) and provide surface variables with 5.5

km spatial resolution from 1984 to 2021.

Precipitation and temperature from weather stations have also been used for the evaluation of HighResClimNevada. Hourly data from the SAIH HidroSur (SAIH-S, http://www.redhidrosurmedioambiente.es) and SAIH Guadalquivir (SAIH-G, https://www.chguadalquivir.es), which are networks of high-quality stations with hourly precipitation and temperature data from 1997 and 2001, respectively, to the present, were used. To preserve as much quality climate data as possible, SAIH-G

and SAIH-S time series with at least 85% of records for a period of at least 19 years within the study period were used. For



precipitation, the SAIH-S and SAIH-G stations were 19 and 13, respectively. For temperature, however, two stations were selected and only from SAIH-S.

All of these databases were downloaded at their native temporal resolution for the time period covered by HighResClimNevada and then aggregated to daily scale in cases where the resolution differed.

**Table 2: Precipitation and temperature datasets used as references in the HighResClimNevada evaluation, classified according to their origin. Daily maximum, mean, and minimum precipitation and temperatures are represented by pr, tasmax, tasmean, and tasmin, respectively. The subdaily temperature is denoted as ta in order to differentiate with daily values.**

| Name | Coverage (resolution / number of stations) | Temporal frequency | Available (analysis) period | Variable |
|---|---|---|---|---|
| **Gridded dataset developed with in-situ stations** | | | | |
| ROCIO_IBEB | Peninsular Spain and Balearic Islands (0.05º) | daily | 1951-2022 (1991-2022) | pr, tasmax, tasmin |
| UGR-SNGrid | Sierra Nevada (200 m) | monthly | 1990-2020 (1991-2020) | pr |
| **Satellite-derived gridded products** | | | | |
| CMORPH | Latitudinal bands from 60ºN and 60º S (8 km) | half-hourly | 1998-present (1998-2022) | pr |
| GPM IMERG | Quasi global (0.1º) | half-hourly | 2000-2021 (2001-2020) | pr |
| CHIRPS | Latitudinal bands from 50ºN to 50º S (0.05º) | daily | 1981-present (1991-2022) | pr |
| **Reanalysis gridded products** | | | | |
| ERA5-Land | Global (~9km) | 1-hourly | 1950-present (1991-2022) | pr, ta |
| CERRA | Europe (5.5 km) | 3-hourly | 1984-2021 (1991-2021) | ta |
| CERRA-Land | Europe (5.5 km) | daily | 1984-2021 (1991-2021) | pr |
| **Punctual in-situ stations** | | | | |
| SAIH stations (Automatic Hydrological Information System) | Guadalquivir (SAIH-G, 9) and southern (SAIH-S, 13) hydrological basins | 1-hourly | SAIH-G: 2001-present (2001-2022), SAIH-S: 1997-present (1997-2022) | pr, ta |

## 2.4. Climate variables in HighResClimNevada: format and file organization

HighResClimNevada has been structured in netCDF files that contain 3D climate fields (time x latitude x longitude) covering
longitudes from 3.85ºW to 2.40ºW and latitudes from 36.50ºN to 37.50ºN (see Fig. 1) over a period from January 1991 to December 2022. All files also provide a 2D mesh with the altitude (z) at sea level expressed in meters above the mean sea level. The database is divided into (1) primary climate variables, (2) bioclimatic variables, (3) extreme ETCCDI climate indices, and (4) precipitation-hour extreme variables.



### 2.4.4. Primary climate variables

-  Near-surface temperature (ta, ºC): temperature plays a crucial role in ecosystem functions, being variables commonly used to describe the climate in a region. Near surface (2 m) temperature was obtained from raw simulation outputs with a 10-minute temporal resolution. Then, hourly (ta) and daily maximum (tasmax), mean (tasmean), and minimum (tasmin) temperatures were calculated based on these values.

  -  Precipitation (pr, kg m$^{-2}$): the accumulated precipitation amounts were obtained from raw 10-minute WRF. As for
temperature, these values were then aggregated on hourly (mm/hour) and daily (mm/day) scales to be part of the HighResClimNevada datasets.

  -  Near-surface relative humidity (hur, %): changes in humidity over land also have important implications on ecosystems. 3-hourly outputs of water vapor mixing ratio at 2 m (r, kg kg$^{-1}$), near surface temperature (ta, ºC), and surface pressure (ps, hPa) provided by WRF were used to estimate hur using Equations 1, 2, and 3.

$$e_s = 6.11 \; 10^{\frac{7.5(ta - 273.15)}{(ta - 273.15) + 237.3}} \tag{1}$$

$$r_s = \frac{0.622 e_s}{ps - e_s} \tag{2}$$

$$hur = \frac{r}{r_s} \times 100 \tag{3}$$

where $e_s$ is the water vapor pressure and $r_s$ is the saturated mixing ratio in kg kg$^1$. Then, 3-hourly hur were used to estimate the daily averages.

  -  Surface pressure (ps, Pa): atmospheric pressure was averaged at a daily scale using the 3-hourly surface pressure WRF outputs.

  -  Surface net radiation (net_radiation, W m$^{-2}$): it is the radiant energy available at the surface to perform work inside the
ecosystem and is critical for maintaining biological and physical processes. Surface net radiation is calculated as the balance between the absorbed, reflected, and emitted energy by the Earth's surface, using 3-hourly outputs and following Equation 4.

$$\text{net\_radiation} = \left( R_S\downarrow - R_S\uparrow \right) + \left( R_L\downarrow - R_L\uparrow \right) \tag{4}$$

where $R_s\downarrow$ (W m$^{-2}$) is the incoming solar radiation, $R_s\uparrow$ (W m$^{-2}$), is the proportion of the solar radiation reflected by the surface, $R_L\downarrow$ (W m$^{-2}$) is the incoming longwave radiation emitted by the atmosphere, and $R_L\uparrow$ (W m$^{-2}$) is the radiation
emitted by the surface.

  -  Surface wind speed (wind_speed, ms$^{-1}$): the daily 10-meter wind speed was calculated using 3-hour outputs of U10 (m/s) and V10 (m/s), which are the eastward and northward WRF wind components at 10 m, respectively, using Equation 5.

$$\text{wind\_speed} = \sqrt{U10^2 + V10^2} \tag{5}$$



### 2.4.5. Bioclimatic variables

Based on monthly temperature values and precipitation, bioclimatic variables have been calculated and are available at
an annual scale in the HighResClimNevada database. These indices, which describe the ecological and environmental
status of ecosystems, were obtained following BIOCLIM WorldClim (Fick and Hijmans, 2017; Hijmans et al., 2005),
CHELSA Bioclim (Karger et al., 2017), and CMCC-BioClimInd (Noce et al., 2020) standard definitions. Here, we
considered those bioclimatic variables with interest for semi-arid mountains and due to their relevance in agricultural and
ecological applications, which are:

- Annual mean temperature (BIO1, ºC): it is calculated for each year by averaging tasmean. Similarly, seasonal values
  have been considered, with summer (June-July-August, JJA) and winter (December-January-February, DJF)
  representing the dry/warm and wet/cold seasons, respectively. To reflect intermediate seasons, this variable was also
  considered for spring (March-April-May, MAM) and autumn (September-October-November, SON). Additionally,
  the annual average of tasmax ($BIO1_{max}$) and tasmin ($BIO1_{min}$) were calculated as well as the subsequent values for
  each season.

- Mean Diurnal Range (DTR or BIO2, ºC): it is the difference between the monthly tasmax and tasmin. In this work,
  BIO2 is referred to as DTR as defined by the ETCCDI as an extreme index of temperature. That is, the daily difference
  between tasmax and tasmin was calculated to obtain annual means of DTR.

- Isothermality (BIO3, %): it measures the degree of daily temperature fluctuations (BIO2) compared to annual
  variations throughout extreme months (BIO7) according to Equation 6.

$$BIO3 = \frac{BIO2}{BIO7} \text{ x } 100 \tag{6}$$

- Temperature seasonality (BIO4, ºC): it refers to temperature fluctuations throughout the year. BIO4 is computed as
  the standard deviation of monthly tasmean, which is obtained using the twelve-monthly values from each year (Noce
  et al., 2020).

- Maximum temperature of warmest month (BIO5, ºC): it is defined through the monthly means of tasmax. Thus, for
  each year, the monthly tasmax value of the month with the highest temperature is selected as the maximum
  temperature of warmest month of that year.

- Minimum temperature of coldest month (BIO6, ºC): similarly to BIO5, monthly means of tasmin are computed
  throughout the years, and then minimum values for each year are taken and the minimum temperature of coldest
  month.

- Annual temperature range (BIO7, ºC): it reflects the range of temperatures between the coldest and the warmest
  months.

$$BIO7 = \ BIO5 - BIO6 \tag{7}$$



- Mean temperature of wettest quarter (BIO8, ºC): wettest quarters were obtained using as a reference the spatial mean of HighResClimNevada pr in the SN Natural Park. For each year, the 3-month precipitation moving sum is calculated, and the maximum value is determined through its central month. BIO8 is then calculated as the average tasmean of the three consecutive averages starting in the wettest quarters (Noce et al., 2020).

- Mean temperature of driest quarter (BIO9, ºC): similarly to BIO8, driest quarter is obtained for each year looking for the minimum 3-month moving sum determined through its central month. Then, mean temperatures for each year are computed through the 3 consecutive months, starting in the driest quarters.

- Annual precipitation (BIO12, kg m$^{-2}$): the total amount of precipitation expressed in mm per year was computed as the average of the annual sum of precipitation. Similarly to temperature, seasonal values (BIO12$_{XXX}$ with XXX being DJF, MAM, JJA, and SON) were obtained.

- Precipitation of wettest month (BIO13, kg m$^{-2}$): similarly to BIO5, it is computed as the highest monthly sum of precipitation for each year.

- Precipitation seasonality (BIO15, %): it shows the precipitation variations throughout the year and is calculated as the coefficient of variation of monthly values in each year (Equation 8), expressed as percentage.

$$BIO15 = \frac{s_{pr}}{\overline{pr}} \times 100 \tag{8}$$

where $s_{pr}$ is the annual standard deviation and $\overline{pr}$ is the mean.

- Precipitation of coldest quarter (BIO19, kg m$^{-2}$): this metric is considered since SN serves as a "water tower" for neighboring regions. Similarly to BIO8 and BIO9, coldest quarters of each year are determined as the lowest temperature of the 3-month moving average from HighResClimNevada. For each year, the precipitation amount is then calculated for the three consecutive months starting in the coldest quarters.

### 2.4.6. Extreme ETCCDI climate indices

Temperature and precipitation extremes indices from the Expert Team on Climate Change, Detection, and Indices were also calculated. Here, extreme variables with relevance in mountain ecosystems have been considered, and these are:

- Warmest day (TXx, ºC) and coldest night (TNn, ºC): TXx represents the maximum value of tasmax in the warmest month. In the same way, TNn is understood as the lowest value of tasmin in the coldest month.

- Growing season length (GSL, days): the growth season is the time of year when plants grow effectively. According to the ETCCDI, GSL is defined as the number of days per year between the first occurrence of at least 6 days with tasmean > 5 ºC and the first occurrence (after July 1$^{st}$) of 6 days with tasmean < 5 °C.

- Icing days (ID, days) and frost days (TNltm2, days): here, mountainous cold extreme patterns are represented by the icing days (annual number of days with tasmax < 0 ºC) and frost days* (annual number of days with tasmin < -2 ºC). Note that the original definition for frost day has a threshold of tasmin < 0 ºC, but SN high mountain region, a more extreme threshold must be considered.



- Wet, heavy, and very heavy precipitation days (R1mm, R10mm, and R20mm, expressed in days/year): the frequency of extreme precipitation is explored through the number of days per year with pr > 1 mm (wet days), pr > 10 mm (heavy precipitation days), and pr > 20 mm (very heavy precipitation days).

- Simple daily intensity index (SDII, kg m$^{-2}$) and wettest pentad (rx5day, expressed in kg m$^{-2}$): SDII calculates the mean annual pr when pr > 1 mm and rx5day is the highest total quantity of pr falling on 5 consecutive days. These indices are calculated as a proxy for the magnitude of daily extreme precipitation.

- Consecutive dry days (CDD, expressed in days/year): longest period in a year of consecutive days with less than 1 mm of pr per day. The significance of this index lies in the fact that it serves as a drought indicator, a very relevant aspect in a semiarid region such as SN.

### 2.4.7. Precipitation-hour extreme indices

Finally, very extreme precipitation events were also considered through precipitation-hour indices, which are:

- Wet-hour frequency (F$_{wet-hour}$,%): percentage of hours per year corresponding to wet hours (pr > 0.1 mm).

- Wet-hour Intensity (I$_{wet-hour}$, kg m$^{-2}$): refers to the average amount of precipitation in one hour only considering wet hours (pr > 0.1 mm).

- Maximum amount of precipitation in the wettest month (PRWw, kg m$^{-2}$): following the philosophy of ETCCDI extremes such as TXx or TNn, PRWw is calculated by determining the maximum amount of precipitation in each month, and then the maximum value is determined for each year to obtain the PRWw time series.



**Table 1: Bioclimatic variables, ETCCDI, and precipitation-hour extreme indices available in HighResClimNevada. Asterisks (\*) indicate self-defined variables, which are based on original bioclimatic variables or ETCCDI indices. The table includes the name of the index, the variables involved and time aggregation used for this calculation, the acronym for the index, and the units in which this is expressed.**

| Name | Variable (time aggregation) | Acronym | Units |
|---|---|---|---|
| **WorldClim bioclimatic variables** | | | |
| Annual mean temperature | tasmean (monthly) | BIO1 | ºC |
| Annual means of maximum temperature* | tasmax (monthly) | BIO1max | ºC |
| Annual means of minimum temperature* | tasmin (monthly) | BIO1min | ºC |
| Seasonal mean temperature* | tasmean (monthly) | BIO1 | ºC |
| Seasonal mean of maximum temperature* | tasmax (monthly) | BIO1max | ºC |
| Seasonal mean of minimum temperature* | tasmin (monthly) | BIO1min | ºC |
| Isothermality | tasmean (monthly) | BIO3 | % |
| Temperature seasonality | pr (monthly) | BIO4 | ºC |
| Maximum temperature of warmest month | tasmax (monthly) | BIO5 | ºC |
| Minimum temperature of coldest month | tasmin (monthly) | BIO6 | ºC |
| Annual temperature range | tasmax, tasmin (monthly) | BIO7 | ºC |
| Mean temperature in wettest quarter | tasmean (monthly) | BIO8 | ºC |
| Mean temperature in driest quarter | tasmean (monthly) | BIO9 | ºC |
| Annual precipitation | pr (monthly) | BIO12 | kg m$^{-2}$ |
| Seasonal precipitation* | pr (monthly) | BIO12* | kg m$^{-2}$ |
| Precipitation of wettest month | pr (monthly) | BIO13 | kg m$^{-2}$ |
| Precipitation seasonality | pr (monthly) | BIO15 | % |
| Precipitation in the coldest quarter | pr (monthly) | BIO19 | kg m$^{-2}$ |
| | | | |
| **ETCCDI extreme indices** | | | |
| Daily temperature range | tasmax, tasmin (daily) | DTR (BIO2) | ºC |
| warmest day | tasmax (daily) | TXx | ºC |
| coldest night | tasmin (daily) | TNn | ºC |
| Growing season length | tasmean (daily) | GSL | days/year |
| Icing days | tasmax (daily) | ID | days/year |
| Frost days* | tasmin (daily) | TNltm2 | days/year |
| wet days | pr (daily) | R1mm | days/year |
| Heavy precipitation days | pr (daily) | R10mm | days/year |
| Very heavy precipitation days | pr (daily) | R20mm | days/year |
| Simple daily Intensity | pr (daily) | SDII | kg m$^{-2}$ |
| Wettest pentad | pr (daily) | Rx5day | kg m$^{-2}$ |
| Maximum length of dry spell | pr (daily) | CDD | days/year |
| | | | |
| **Precipitation-hour extreme indices** | | | |
| Wet-hour frequency | pr (hourly) | $F_{wet-hour}$ | % |
| Wet-hour intensity | pr (hourly) | $I_{wet-hour}$ | kg m$^{-2}$ |
| Maximum amount in the wettest month | pr (hourly) | PRWw | kg m$^{-2}$ |



## 3.    Results

### 3.1.    Temperature and precipitation distributions

Fig. 2 shows the probability density functions (PDFs) of tasmean (Fig. 2a) and pseudo-PDF of pr (Fig. 2b), as well as the annual cycles of monthly values of both variables (Figs. 2c and 2d) for reference climate datasets and HighResClimNevada. WRF data from the parent domain (05-WRF) have also been included because they are closer in spatial resolution to the reference datasets and hence more comparable. To obtain the PDFs, pseudo-PDFs, and annual cycles, all grid points in the SN Natural Park were considered. PDFs were estimated using a 0.5 ºC bin, whereas pseudo-PDFs were obtained using an approach

similar to that provided in Argüeso et al. (2012). Therefore, we calculated the amount of precipitation in millimeters (mm) contributed by daily events, or bins, using 2 mm/day bins for daily precipitation greater than 0.1 mm.

In general, HighResClimNevada tasmean shows similar PDFs to other climate products (values ranging from -15 to 35 ºC) with certain bimodal features with two peaks of frequency (the highest peak around 5 ºC, while the lighter one is around 20 ºC). The major difference in the shape of the distributions appears in ROCIO_IBEB, where the bimodal character is more

diffuse with its second mode shifted towards higher values. In terms of precipitation (Fig. 2b), differences between pseudo-PDFs are shown depending on the spatial resolution and the nature of the data. Events with pr around 10 mm/day seem to contribute to the highest amount of precipitation in HighResClimNevada. CHIRPS shows a similar shape in its pseudo-PDF, but slightly shifted towards higher precipitation events. For the other climatic products, however, the highest amount of pr occurs for lighter events (5 mm/day or less), which is more pronounced in GPM IMERG and ERA5-Land. The latter suggests

that the shape of the PDF is influenced, at least in part, by the spatial resolution of the data.

HighResClimNevada exhibits an annual cycle of monthly tasmean similar to 05-WRF and ROCIO_IBEB (Fig. 2c). However, underestimations of roughly 3 ºC are shown compared to ERA5-Land and slight overestimations when CERRA is employed as a reference. In any case, all climatic products reveal an annual cycle with the highest temperature in July and the lowest in January. Concerning the spatiotemporal distribution of temperature, the violin plot in Fig. 2c shows that HighResClimNevada,

05-WRF, and CERRA are more widespread, which could be related to the spatial resolution (the higher the spatial resolution, the greater the dispersion) and the nature of the data, as these three climatic products are derived from modeled data. It should be noted that in high mountain regions, the number of weather stations is limited, making temperature characterization more difficult for station-based products such as ROCIO_IBEB. Regarding precipitation, all climate products have a similar shape in their annual cycle (Fig. 2d), with the highest precipitation in March and the lowest during July. However, the amount of

precipitation differs month by month. HighResClimNevada produces a high amount of pr (726 mm/year), second only to CHIRPS (792 mm/year) (inner figure in Fig. 2d), which is due to a higher amount of pr throughout the autumn and winter months. During the summer, however, HighResClimNevada appears to show a lower pr amount than reanalysis-based data like CERRA-LAND or ERA5-Land.





**Figure 2: (a)** Probability density functions (PDFs) of daily mean temperature (tasmean) and **(b)** pseudo-PDFs of the daily precipitation (pr) for all climatic products. Annual cycle of the monthly mean **(c)** temperature and **(d)** precipitation. The inner figure at c shows a violin plot with the monthly mean temperature values at each grid point and for each database. The inner figure at d shows a bar chart with the average annual precipitation amount (expressed in millimeters per year) in each database.



### 3.2. Temperature and precipitation bioclimatic variables and extremes

Fig. 3 depicts the annual average of tasmax (BIO1$_{max}$), tasmean (BIO1), and tasmin (BIO1$_{min}$) for reference datasets (ERA5-Land, ROCIO_IBEB, and CERRA) and modeled data (05-WRF and HighResClimNevada). The dots in each panel represent the temperature recorded by SAIH stations. All climatic products are represented in their native spatial resolution to avoid problems caused by the interpolation method and the difference between spatial resolutions. Additionally, the temporal evolution of temperature in the Natural Park is depicted as warming stripes or standardized anomalies (i.e., annual temperature minus the annual mean in the common period, 2001-2020, divided by its standard deviation) under each map. Blue colors represent cooler than average years, while red colors suggest warmer than normal years. Blue and red intensities represent the magnitude of the anomaly. BIO1 characterizes the amount of energy captured by an ecosystem during a year, making its knowledge highly relevant for the conservation of high mountain ecosystems. In general, results show that temperature values (BIO1$_{max}$, BIO1, and BIO1$_{min}$) are very influenced by elevation, with the lowest values over the highest mountain peaks. All climatic products exhibit both similar spatial patterns and interannual variability, the latter as shown by warming stripes. These results, moreover, evidence the effect of spatial resolution. That is, datasets with coarser spatial resolution show more homogeneous temperature patterns, as they are not able to capture the elevation effects adequately. As a result, temperature in the Mulhacen and Veleta peaks in these datasets is usually higher (e.g., while ERA5-Land shows BIO1$_{min}$ values around 7 ºC in the highest altitudes, HighResClimNevada indicates values around -2 ºC). Moreover, HighResClimNevada produces comparable values to those from SAIH stations, as observed in 05-WRF, CERRA, and ROCIO_IBEB. However, ERA5-Land tends to overestimate both BIO1$_{max}$, BIO1, and BIO1$_{min}$. Comparable results can be found for seasonal values (Figs. S1-S4 in the supplementary figures).

HighResClimNevada, as other climatic products, accurately captures the main spatial patterns of annual precipitation in SN, showing an east-southwest gradient (Fig. 4). However, when we focus on high elevation areas, differences emerge that are determined by the spatial resolution and the nature of the data. Thus, in this region, downscaled fields (i.e., HighResClimNevada and 05-WRF), high-resolution satellite-derived products (CHIRPS), and high-resolution reanalysis (CERRA-Land) exhibit higher annual precipitation (greater than 1100 mm) than station-based data (ROCIO_IBEB and UGR-SNGrid), and others satellite-based and reanalysis data with lower resolution (GPM IMERG and ERA5-Land), where annual precipitation ranged from 550 to 700 mm approximately. These findings suggest the importance of spatial resolution in representing this variable, as datasets with resolutions of 5 km (or less) show a significant peak in annual cumulative precipitation at higher altitude. Lower-resolution datasets, on the other hand, exhibit more homogeneous precipitation patterns. This behavior is also shown in UGR-SNGrid and ROCIO_IBEB, where stations at high altitudes for precipitation calculation are scarce. In comparison to the SAIH stations, HighResClimNevada shows higher BIO12, particularly at high altitude, where UGR-SNGrid produces closer values to these stations. It should be noted that the SAIH stations were one of the data sources in the development of UGR-SNGrid; therefore, similarities between the two are expected. It is also important to highlight the challenge of maintaining high mountain weather stations due to their limited accessibility, especially during storm events when

observed values are likely to be lower than those actually occurring. As a result, we know that, while these stations are reasonably good, their geographical position limits their quality. Grids based on these values are so likely to inherit this type of problem. HighResClimNevada presents a general agreement with other climatic products in the interannual variability of precipitation, as shown by the drying stripes. Similar conclusions can be drawn regarding the amount of precipitation that falls on average during winter (Fig. S5 in the supplementary material) and spring (Fig. S6). For summer (Fig. S7), all climatic products show precipitation values lower than 75 mm except for CERRA-Land and SAIH stations, which have precipitation values above 125, although in different locations. For autumn (Fig. S8), WRF outputs and CERRA-Land again show pr above 300 mm in the northwest, which is not replicated by other climatic products. However, for this season, the spatial pattern across the National Park appears to be more accurate than in other climatic products when compared to SAIH stations.

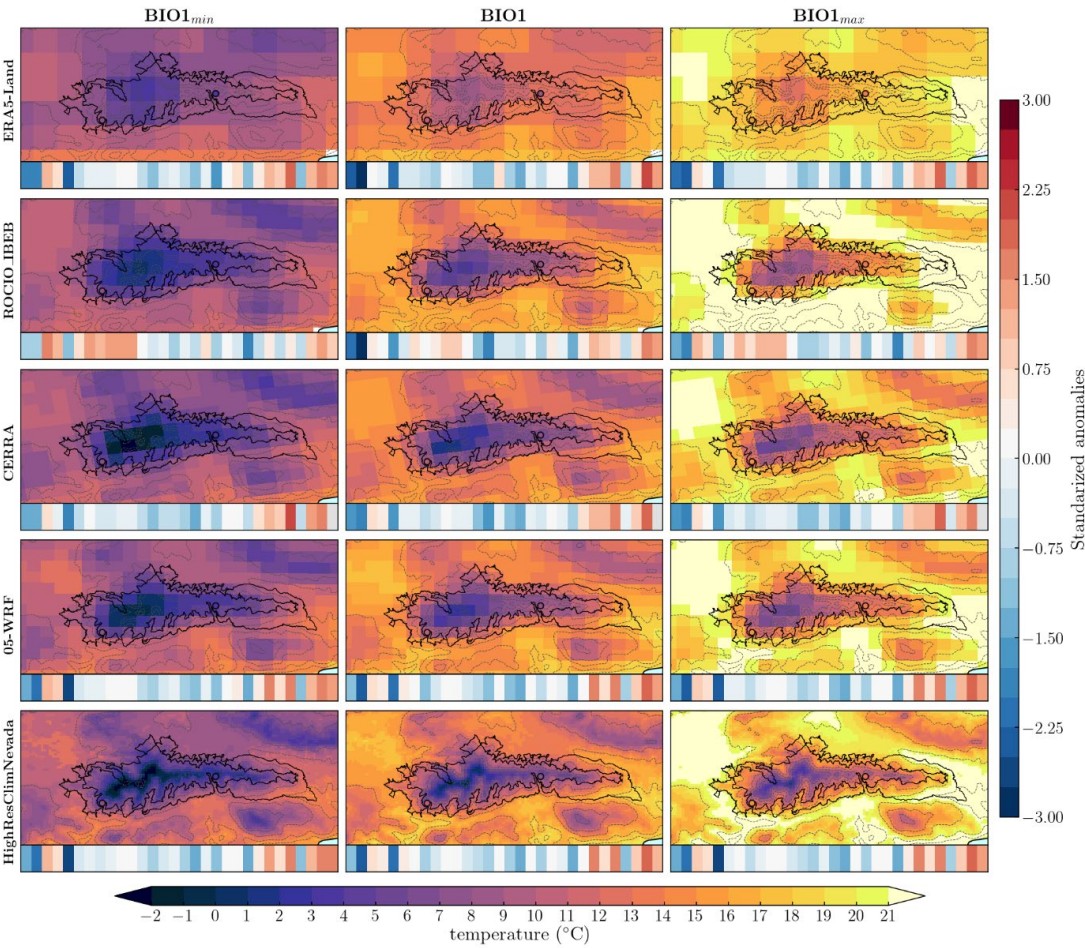

**Figure 3: Annual mean of the daily minimum, mean, and maximum temperatures (BIO1$_{min}$, BIO1, and BIO1$_{max}$) for each reference dataset (ERA5-Land, ROCIO_IBEB, and CERRA) as well as for downscaled data (05-WRF and HighResClimNevada). The dots in each panel represent the temperature recorded by SAIH stations, and the black solid lines display the National and Natural Park boundaries. Normalized anomalies for SN Natural Park over the period 2001-2020 for each database are shown below the corresponding spatial map in the form of warming stripes. In this representation, red colors indicate positive standardized anomalies and blue colours negative values compared to the mean value for the whole period.**





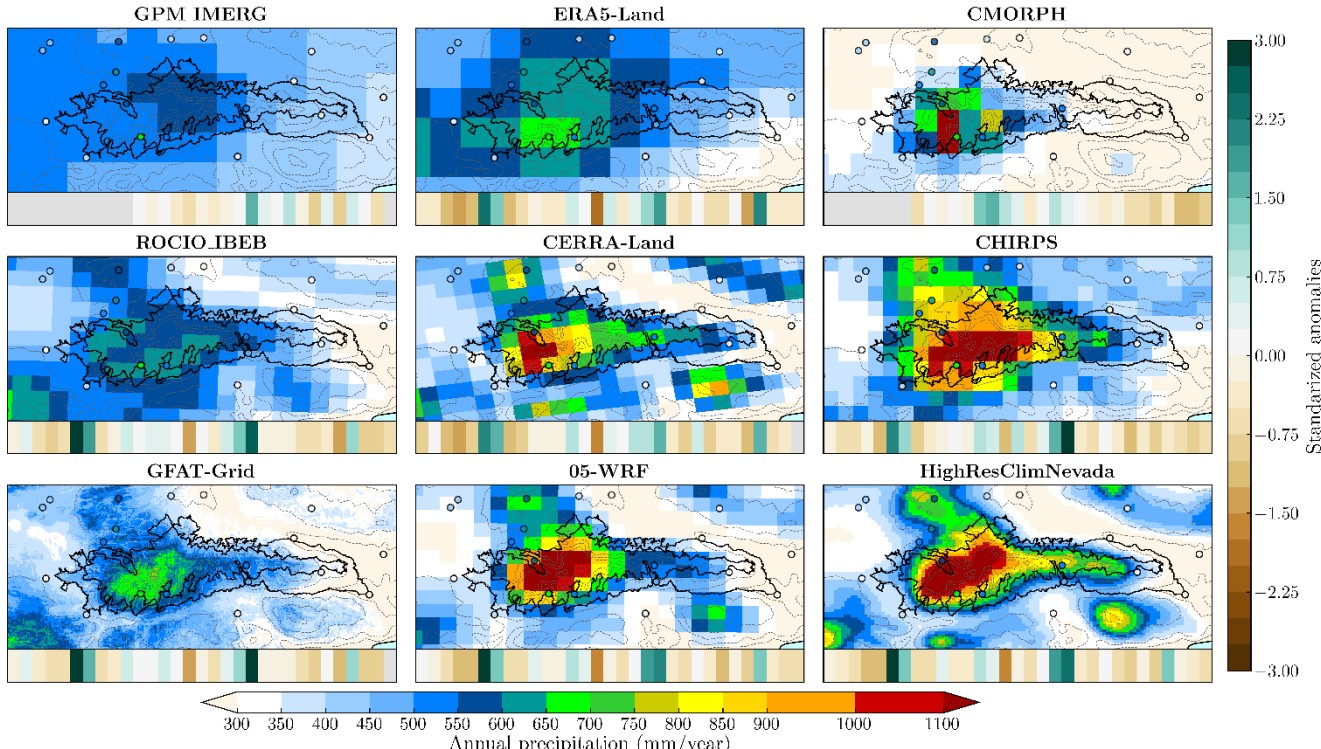

**Figure 4: Average of the annual precipitation (BIO12, mm/year) for station-based data (ROCIO_IBEB and UGR-SNGrid), satellite image-based data (GPM IMERG, CMORPH, and CHIRPS), reanalysis data (ERA5-Land and CERRA-Land), and downscaled precipitation data (05-WRF and HighResClimNevada). The dots in each panel represent the temperature recorded by SAIH stations and the black solid lines display the National and Natural Park boundaries. The annual standardized anomalies over the period 2001-2020 for each database are shown below the corresponding spatial map in the form of drying stripes. In this representation, green colors indicate wet anomalies and brown colors the negative ones. Gray colors indicate no data for that year.**

Fig. 5 depicts the long-term mean spatial patterns of temperature bioclimatic variables (Figs. from 5a to 5f) and ETCCDI extreme indices (Figs. 5g to 5l) for HighResClimNevada. In each panel, dots represent the corresponding values for SAIH stations. In general, HighResClimNevada shows comparable behavior to other climatic products (Figs. S9 to S12) and stations, but with an apparent improvement due to the enhanced resolution. That is, HighResClimNevada characterizes orography-related effects that are not well captured by coarser climatic products like ERA5-Land (Fig. S9). In this regard, elevation influences most of the bioclimatic variables and extremes except for BIO4 and BIO7. Isothermality (BIO3, Fig. 5a) is a bioclimatic measure that compares variability within the day (BIO2 or DTR, Fig. 5g) to variability throughout the year (BIO7, Fig. 5c). BIO3 is crucial for ecosystems, as many species might suffer damage in their population distribution as a result of daily and annual oscillations (O'Donnell and Ignizio, 2012). Values above 100% indicate that daily temperature variability is equal to the annual variability. According to HighResClimNevada, SN has low BIO3 (up to 35%), indicating that temperature fluctuations in this region during the day account for no more than 35% of the total variation across the year. The lowest BIO3 are depicted over the highest peaks, where they reach values less than 25%. In this region, DTR is also the lowest, reaching



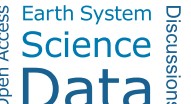

values lower than 6 °C. BIO7, however, reaches values between 23 ºC and 30 ºC, with the maximum located in the northeast
of the National Park of SN. Like BIO7, temperature seasonality (BIO4) illustrates the temperature variability throughout the
year. HighResClimNevada (Fig. 5b), with values between 6 ºC and 7 ºC, has the highest BIO4 values in the northeastern part
of the Natural Park. For this variable, although the results of other climatic products are within the same range, their spatial
patterns differ significantly. BIO5 (Fig. 5d), BIO6 (Fig. 5e), and BIO8 (Fig. 5f) are useful when we are interested in analyzing
if species distributions are affected by warm, wet, and dry temperatures, respectively (O'Donnell and Ignizio, 2012). Although
HighResClimNevada is comparable to other climatic products in terms of these variables (see Figs. S9-S12d, S9-S12e, and
S9-S1f), it appears to show lower values, especially when compared to ERA5-Land and ROCIO_IBEB, higher in the highest
peaks. Similarly, HighResClimNevada has higher TNltm2 (Fig. 5l) and ID (Fig. 5k) than other climatic products, with values
exceeding 160 and 105 days per year, respectively. TNltm2 and ID have critical significance for ecosystems, as prolonged
cold temperatures may cause considerable damage, affecting the survival of plant and animal species (Liu et al., 2018). GSL
is a key variable in vegetable ecosystems because it influences essential functions including hydrology, nutrient cycling,
productivity, and climate feedback (Barnard et al., 2018). In terms of this variable, HighResClimNevada shows values varying
from 270 days/year in SN Natural Park's outermost parts to values below 90 days/year in the highest mountain peaks (Fig. 5j),
which is very similar to other climatic products.

Fig. 6 depicts the interannual climate variability of bioclimatic variables and temperature extreme ETCCDI indices through
the normalized anomalies in the area delimited by the SN Natural Park and estimated using the mean and standard deviation
during the common period (2001-2020). For each index, the four databases, i.e., the three reference temperature datasets and
HighResClimNevada, are plotted according to the order defined by the legend, with the color in each triangle representing the
normalized anomaly for that year. Triangles colored in shades of red indicate that the index takes a value above normal, and
blue is the opposite pattern. In this figure, black triangles indicate that there are no values in that dataset for that year. Overall,
HighResClimNevada has a very similar temporal evolution to other climate products in terms of bioclimatic variables (from
BIO3 to BIO9), suggesting that it is able to capture the SN temperature interannual variability. Thus, in general, anomalies in
HighResClimNevada coincide in sign with those from other climatic products, but also in magnitude. That is all climatic
products exhibited very high (low) BIO3 anomalies in 1997 (2012), with BIO7 lower (higher) than normal. Note that BIO7 is
the denominator for BIO3. This result indicates a small (high) dispersion in monthly temperature in that year when compared
to the long-term climate, which is corroborated by BIO4. In the same way, BIO5 and BIO6 also agree in all products for 1997
(2012), showing that the maximum temperature of the warmest month was lower (higher) than normal, and the minimum
temperature of the coldest month was higher (lower) than normal. Moreover, these years showed an unusually low (high)
temperature in the driest quarter for that year, as indicated by BIO9. In the same way, temperature extremes are also similar in
all climatic products, suggesting also that HighResClimNevada captures the interannual variability of extreme temperature.

Precipitation bioclimatic variables and extremes for HighResClimNevada are represented in Fig. 7. The dots in each panel
reflect the values achieved by SAIH stations, while the black solid lines display the National and Natural Park boundaries. In
general, the results suggest that orography influences the patterns of bioclimatic indices when the precipitation is implicated.





This is especially shown in the patterns of the precipitation of the wettest month (BIO13, Fig. 7a) and the precipitation of the coldest quarter (BIO19, Fig. 7c). Moreover, both variables show a very similar spatial pattern, with BIO13 values ranging from 90 to 350 mm and BIO19 values from 150 to 750 mm. Precipitation seasonality (BIO15, Fig. 7b), with values ranging from 95 to 120%, however, appears not to be influenced by orography, showing the lowest values in the north-northwest part of the SN National Park. Extreme ETCCDI indices are also influenced by orography (from Fig. 7d to Fig. 7i). That is, these indices show the highest extreme precipitation at high altitude in the western mountains and the lowest in the outermost Natural Park's regions. Thus, SN is characterized by R1mm (Fig. 7d), on average, between 35 and 80 rainy days per year, SDII values (Fig. 7g) ranging from 8 and 18 mm/day, and Rx5day (Fig. 7h) with values between 30- and 140 mm per 5 days. Heavy (R10mm, Fig. 7e) and very heavy (R20mm, Fig. 7f) precipitation days, on the other hand, exhibit similar regional patterns to R1mm, with maximum values exceeding 36 and 26 days per year, respectively. In terms of dry spells (CDD, Fig. 7i), HighResClimNevada indicates that the SN Natural Park is affected by 50 to 120 consecutive dry days. In terms of very extreme precipitation values (from Fig. 7j to Fig. 7l), which are based on hourly precipitation, HighResClimNevada shows that wet hours (hourly pr > 0.1 mm) account for 11% of the annual hours at the highest peaks, with values falling below 2% in the southeastern part of the Sierra Nevada National Park (Fig. 7j). The average intensity in wet hours appears to be around 1.6 mm/hour (Fig. 7k), with extreme hourly precipitation up to 25 mm/hour (Fig. 7l). For these latter indices, the orography has an unclear effect, with the highest values in the west and the south.

When the temporal evolution of precipitation values from HighResClimNevada is compared with other climatic products (Fig. 8), we can note more discrepancies than for temperature in the sign of the anomalies. Such discrepancies are greater when HighResClimNevada is compared to CMORPH, but the latter also disagrees with other climatic products. For example, in 2010, a year characterized by high precipitation records in the southern part of IP during the winter and spring, strong positive anomalies in BIO19 (i.e., precipitation of the coldest quarter) are depicted except for CMORPH, which indicates normal conditions for that year. Similarly, positive anomalies are observed for BIO13 (i.e., precipitation of the wettest month), but they are lighter than for BIO19. In terms of BIO15 (i.e., precipitation seasonality), however, there is not a clear consensus between datasets for that year. Very extreme precipitation indices indicate that HighResClimNevada performs well according to reference datasets providing hourly data (i.e., GPM IMERG, ERA5-Land, and CMORPH).

## 4. Data and code availability

Codes, supplementary figure and HighResClimNevada data are available in García-Valdecasas Ojeda et al. (2024) (https://doi.org/10.5281/zenodo.14052394). In this link, daily values of temperature (mean, maximum, and minimum), precipitation, near-surface relative humidity, surface pressure, surface net radiation, and wind speed for the region of Sierra Nevada at 1 km spatial resolution can be found. Additionally, hourly precipitation and temperature datasets are provided as a part of this database as well as bioclimatic variables and extremes. For the evaluation of our HighResClimNevada, different reference data were used, all freely available. AEMET 5km gridded datasets (ROCIO_IBEB version 2) are online available at





https://www.aemet.es/es/serviciosclimaticos, last access: January 2024. UGR-SNGrid is online available at the institutional
repository of the Universidad de Granada (https://hdl.handle.net/10481/95487). GPM IMERG
(https://disc.gsfc.nasa.gov/datasets/GPM_3IMERGHH_07/summary?keywords=%22IMERG%20final%22, last access:
January 2024) and CMORPH rainfall estimations (https://www.ncei.noaa.gov/data/cmorph-high-resolution-global-
precipitation-estimates, last access: January 2024) are freely available online. The Climate Hazards Center of the University
of California Santa Barbara provides the CHIRPS data v2 (https://data.chc.ucsb.edu/products/CHIRPS-2.0, last access: 20
September 2023) for different domains, formats, and resolutions. Here, the global daily version at 0.05º of spatial resolution
was downloaded in netCDF format. CERRA, CERRA-Land, and ERA5-Land are available in the Copernicus Climate Data
Store (CDS) (https://cds.climate.copernicus.eu/, last access: 20 September 2023). The weather station data for SN was
provided by the SAIH networks, and they are available upon request.

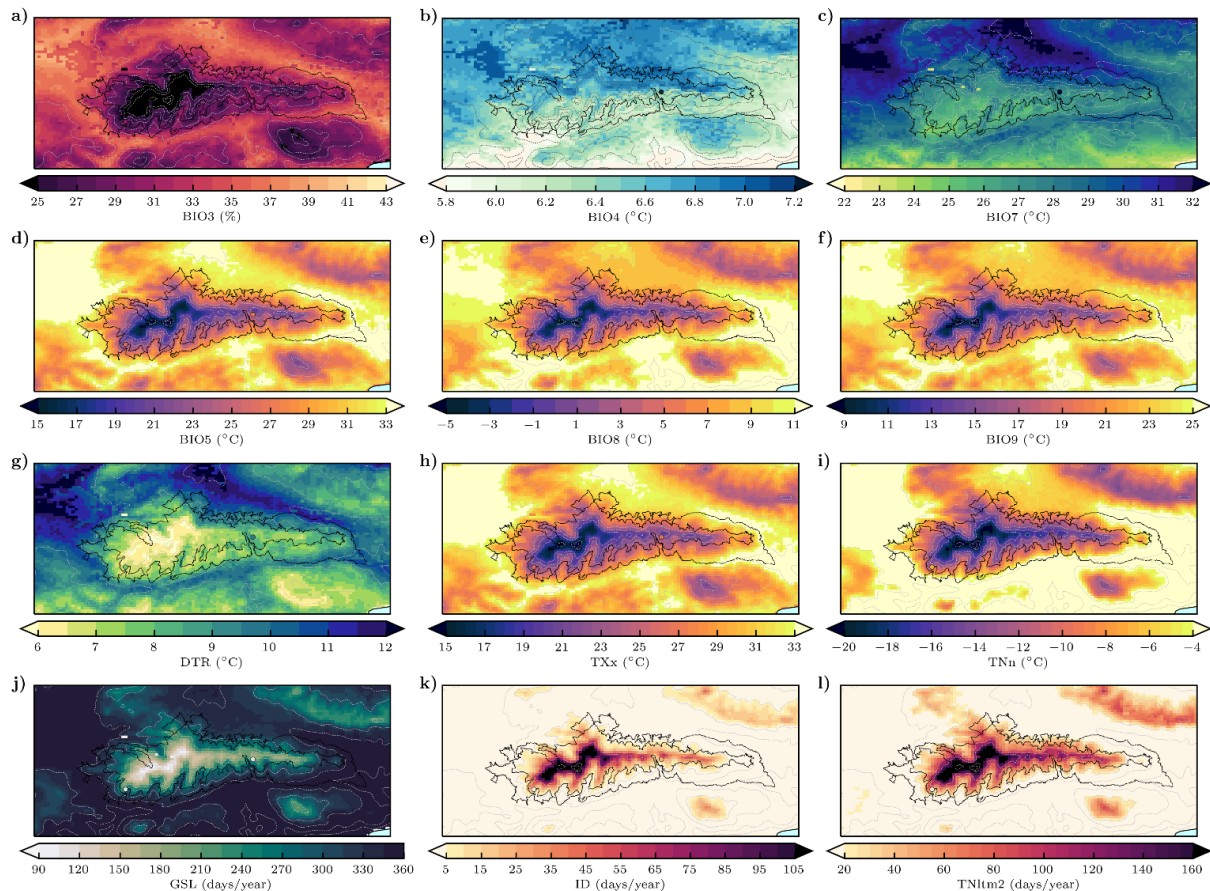

**Figure 5: Spatial patterns of temperature bioclimatic variables and extreme ETCCDI indices in HighResClimNevada. (a) Isothermality (BIO3,%), (b) temperature seasonality (BIO4, ºC), (c) annual temperature range (BIO7, ºC), (d) maximum temperature of warmest month (BIO5, ºC), (e) mean temperature of wettest quarter (BIO8, ºC), (f) mean temperature of driest quarter (BIO9, ºC), (g) daily temperature range (DTR, ºC), (h) maximum warmest day (TXx, ºC), (i) minimum coldest night (TNn, ºC), (j) growing season length (GSL, days/year), (k) icing days (ID, days/year), and (l) frost days* (TNltm2, days/year). The dots in each panel reflect the values achieved by SAIH stations, while the black solid lines display the National and Natural Park boundaries.**



Earth System
Science
Data

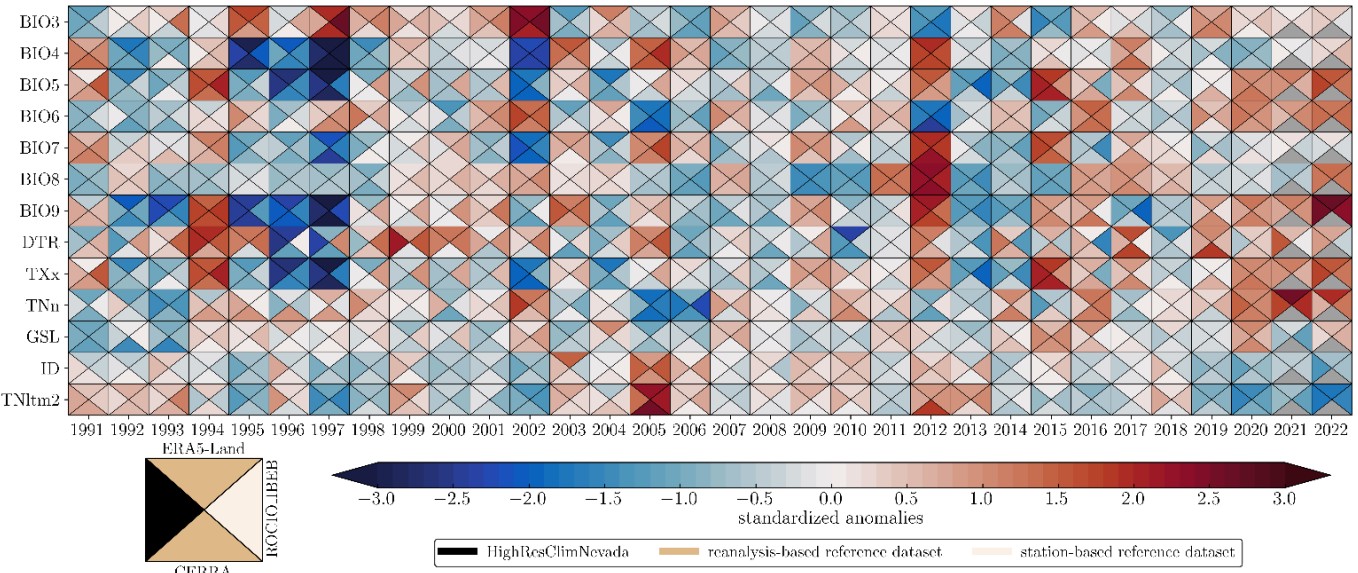

**Figure 6: Normalized temperature anomalies for bioclimatic variables and ETCCDI extreme indices. Anomalies are calculated using**
**the mean and standard deviation in a common period (1991-2022) of each database and spatially averaged across the SN Natural**
**Park. Black triangles indicate that there is no value for that year in the specified database. The legend depicts the sequence of the**
**data in each square (in black, HighResClimNevada), with the nature of each database marked by color.**

## 5.    Discussion and conclusions

We present HighResClimNevada, a climatological 1-km dataset derived from regional climate modeling. To do that, the
Weather Research and Forecasting model v4.3.3 with a configuration especially designed for Sierra Nevada has been used.
SN is a region of special interest due to its high ecological value where information regularly in space at an adequate resolution
is scarce. HighResClimNevada provides hourly and daily primary climate variables structured in netCDF files with variables
stored in a 1-km grid from January 1991 to December 2022. Additionally, and due to the special relevance of this region to
study the impacts of climate change, bioclimatic variables, extreme ETCCDI climate indices, as well as hour-precipitation
extreme indices derived from hourly precipitation were postprocessed and are also part of this climatic product.

HighResClimNevada was compared with reference datasets in order to evaluate its climate performance. These comparisons
showed that HighResClimNevada is a valuable tool for assessing trends in temperature variables, both in terms of extreme and
mean values, as its performance is similar to that of other products. However, due to its higher resolution, it shows a level of
spatial detail that is not available in the other products, which is physically consistent (due to the nature of the product itself)
and necessary for ecological analysis.

In terms of precipitation, more discrepancies are shown, although it is important to note the difficulty in obtaining data in this
complex region. In fact, it is well known that datasets based on observations often have problems in high mountain regions
due to the challenge of maintaining station networks with an adequate density of stations. Moreover, satellite products like

GPM IMERG or CMORPH also show problems characterizing the precipitation amount fallen over mountains, leading to

underestimations (Derin and Yilmaz, 2014; Kazamias et al., 2022; Navarro et al., 2020; Tapiador et al., 2020). However, HighResClimNevada shows a similar behavior in many extreme values to reanalysis products such as CERRA-Land and CHIRPS.

The HighResClimNevada dataset presented here serves as a scientific basis for assessing the impacts of climate change over SN on many sectors, such as on ecological and hydrological systems.


**Figure 7: Spatial patterns of bioclimatic variables and extreme ETCCDI indices of precipitation in HighResClimNevada. (a) precipitation of wettest month (BIO13, mm/month), (b) precipitation seasonality (BIO15, mm/month), (c) precipitation in the coldest quarter (BIO19, mm/month), (d) wet days (R1mm, days/year), (e) heavy precipitation days (R10mm, days/year), (f) very heavy**
**precipitation days (R20mm, days/year), (g) simple daily intensity index (SDII, mm/year), (h) maximum 5-day precipitation (Rx5day, mm), (i) consecutive dry days (CDD, days/year), (j) wet-hour frequency ($F_{wet-day}$, %), (k) wet-hour intensity ($I_{wet-day}$, mm/hour), and (l) maximum amount in the wettest month (PRWw, mm/hour). The dots in each panel reflect the values achieved by SAIH stations, while the black solid lines display the National and Natural Park boundaries.**

Earth System
Science
Data



**Figure 8: Normalized precipitation anomalies for bioclimatic variables, ETCCDI extreme indices, and precipitation-hour extreme indices. Anomalies are calculated using the mean and standard deviation in a common period (1991-2022) of each database and spatially averaged across the SN Natural Park. Green colours indicate wet anomalies while red colours show drier than normal values. Grey triangles indicate that there is no value for that year in the specified database. The legend depicts the sequence of the data in each square (in black, HighResClimNevada), with the nature of each database marked by colours.**

**Acknowledgement:** This research was financially supported by the project "Plan Complementario de I+D+i en el área de Biodiversidad (PCBIO)" funded by the European Union within the framework of the Recovery, Transformation and Resilience Plan - NextGenerationEU and by the Regional Government of Andalucia, the project PID2021-126401OB-I00, funded by MICIU/AEI/10.13039/501100011033 and by FEDER, UE; and LifeWatch-2019-10-UGR-01 co-funded by the Ministry of Science and Innovation through the FEDER funds from the Spanish Pluriregional Operational Program 2014–2020 (POPE)





LifeWatch-ERIC action line; and P20_00035 funded by FEDER/Junta de Andalucía-Consejería de Transformación Económica, Industria, Conocimiento y Universidades.

**Competing interests:** The contact author has declared that none of the authors has any competing interests.

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
