# Peer review of "HighResClimNevada: a high-resolution climatological dataset for a high-altitude region in Southern Spain (Sierra Nevada)"

_Earth System Science Data, 2024_

## Referee Comment (RC2)

In this study, the authors present a new high-resolution dataset over Sierra Nevada in the Iberian Peninsula for the period 1991-2022. The dataset was created with the WRF model, and ERA5 reanalysis data provided the initial and boundary conditions. To test the accuracy of the model, many indices were compared against alternative datasets based on observations and satellite data. The reliable performance of the dataset is highlighted throughout the manuscript. This dataset is valuable for the climate research community working on high mountain environments, but also for other areas such as biology or ecology.

The manuscript follows a logical structure and fits into the scope of Earth System Science Data. However, the authors need to address some comments before it is ready for publication.

**Major comments:**

Not related to any section:

- After reading the paper, it is unclear if the variables are provided in the original Lambert grid from WRF, or if they have been interpolated to a regular lon lat grid. That should be clearly stated in the text. That is my concern since wind speed is calculated using equation 5, and both components depend on the grid in which they are provided. For a regular lon lat grid, the winds should be rotated from WRF's Lambert grid.

*Section 2.2.*

- The authors could briefly explain the model sensitivity analysis that led to the selected configuration in WRF. Only the reference is provided in the current version, but the manuscript can improve if more details are provided. For example: Was the analysis based on temperature? Precipitation? Other variables?
- Line 111 mentions the trade-off between suitability and computational resources, but Table 1 provides specific time steps for each domain. WRF can run using an adaptive time step to reduce the simulations' running time. Was the simulation created using that option? If not, is there any reason why the authors did not use it?
- I would consider a 5 km spatial resolution already a convective scale. Thus, I do not see a reason to use the convection parameterization in the first domain of the model. Did the authors test that alternative?

*Section 2.3.*

- Could the authors provide briefly more details about the interpolation method used in the RegRAIN package? That could ease the reading and understanding of that part of the text.

*Sections 2.4.4, 2.4.5, and 2.4.7*

- I am not sure if the units of the variables related to precipitation are correctly stated in these two sections. For example, Precipitation is defined between brackets as kg/m2, but then in the explanation, it is defined as mm/hour. This also happens in BIO12, BIO13, BIO19, Wet-hour Intensity and Maximum amount of precipitation in the wettest month. Could the authors check these mismatches?
- Is the definition of the Simple Daily Intensity Index correct? It is defined as the mean annual pr when pr >1mm, but I would guess that it should be related to daily values instead of annual values. Please check that.

*Section 3.1*

- The manuscript would benefit from a short explanation of the pseudo-PDFs. More concretely, why and how they are calculated.

- Lines 317-320: I missed a comment about the shape of the probability distribution function shown by ERA5-Land compared to the other datasets. I would say that it is also different around 20ºC.

*Section 3.2*

- Line 353: Can the authors elaborate more on what they mean by "amount of energy"? That sentence is referred to BIO1, and that is the annual mean temperature, so it is not an energy.
- Figure 4: There is a mismatch between the labels in the Figure and the Caption. UGR-SNGrid is labelled as GFAT-Grid in the figure. The same happens in Figure S7 in the supplementary.

**Minor comments:**

- References should be listed in chronological order throughout the manuscript (e.g., lines 46,48-49, 68-69, etc).
- Line 76: I would start a new paragraph to explain the structure of the publication.
- Line 110: initial soil moisture conditions or soil moisture initial conditions.
- Line 156: 200 m instead of two hundred.
- Table 2: I printed the PDF and it seems that there is a problem with the formatting of the text related to the coverage of CHIRPS. It appears as bold.
- Table 2: Define ta in the caption, as it is not explained in the text yet. In this version, only pr, tasmax and tasmin are explained in the caption.
- Line 200: $kg*kg^{-1}$
- Lines 238-239: "…, and then minimum values for each year are taken AS the minimum…"

- Line 270: I suggest removing the * as the note is in the following line.
- Page 13: Table 3 is defined as Table 1 again
- Line 334: The highest precipitation is found in December, right?
- Line 397 – Caption Fig. 4: It states temperature, but it should be precipitation recorded by the SAIH stations.
- Line 418: BIO5, BIO8 and BIO9, according to the subfigures mentioned in that line.
- Lines 337-338: Something is missing in the sentence. Otherwise, I suggest rewriting it.
- Line 455: With values between 30 and 140 mm.

---

## Author Comment (AC1)

**Reply to Anonymous Referee #2**

In this study, the authors present a new high-resolution dataset over Sierra Nevada in the Iberian Peninsula for the period 1991-2022. The dataset was created with the WRF model, and ERA5 reanalysis data provided the initial and boundary conditions. To test the accuracy of the model, many indices were compared against alternative datasets based on observations and satellite data. The reliable performance of the dataset is highlighted throughout the manuscript. This dataset is valuable for the climate research community working on high mountain environments, but also for other areas such as biology or ecology.

The manuscript follows a logical structure and fits into the scope of Earth System Science Data. However, the authors need to address some comments before it is ready for publication.

**Reply:** Authors thank the reviewer for reading the manuscript and taking the time to review it, asking critical questions, which will help us to improve the quality of the manuscript. This comment will be therefore included in the acknowledgements section. We have responded to the referee's request point by point indicating the actions to be taken in the final revision of the manuscript. All our replies are included in blue.

**Major comments:**

(1) Not related to any section: - After reading the paper, it is unclear if the variables are provided in the original Lambert grid from WRF, or if they have been interpolated to a regular lon lat grid. That should be clearly stated in the text. That is my concern since wind speed is calculated using equation 5, and both components depend on the grid in which they are provided. For a regular lon lat grid, the winds should be rotated from WRF's Lambert grid.
**Reply:** we are sorry for the misunderstanding. The variables are provided according to the original Mercator grid provided by the model. In this regard we will add a clarification in the new version of the manuscript saying that the data are in their original mesh and therefore it has not been necessary to rotate the wind.

(2) Section 2.2. - The authors could briefly explain the model sensitivity analysis that led to the selected configuration in WRF. Only the reference is provided in the current version, but the manuscript can improve if more details are provided. For example: Was the analysis based on temperature? Precipitation? Other variables? - Line 111 mentions the trade-off between suitability and computational resources, but Table 1 provides specific time steps for each domain. WRF can run using an adaptive time step to reduce the simulations' running time. Was the simulation created using that option? If not, is there any reason why the authors did not use it? - I would consider a 5 km spatial resolution already a convective scale. Thus, I do not see a reason to use the convection parameterization in the first domain of the model. Did the authors test that alternative?
**Reply:** The sensitivity study aimed to analyze the WRF model performance over Andalusia (southern Spain) using 12 1-year simulations resulting from combining different microphysics and cumulus options in the parent domain (d01) (i.e., the cumulus scheme was only switched off in the inner domain). These experiments were evaluated in terms of precipitation and maximum and minimum temperatures using

different data as reference. Among the different options evaluated in this study in relation to the clusters, the option of using convection off in the parent domain (d01) was also tested, as indicated by the reviewer, but, although this was one of the options that obtained better results, in general, the use of Grell Freitas (GF) convection in d01 seemed to yield more adequate results in all the analyzed variables. Regarding this study, as suggested by the reviewer, we will add more details about the sensitivity study performed in the new version of the manuscript.

On the other hand, we use a fixed time-step following the recommendations of the WRF model developers (6*spatial resolution). This type of timestep is widely used in WRF in convection-permitting simulations and has been shown to show adequate results in the IP. However, it could be an aspect to investigate for future studies, thank you for this suggestion. In any case, this type of time step should be used with caution and as indicated in some studies (e.g., De Morais and Guerrero, 2018) seems to be less advisable than the use of fixed values. In addition, it seems that in long simulations there can be marked differences between the results of using one or the other type of time step, an aspect not solved at least until version 4.2 (see the https://forum.mmm.ucar.edu/threads/different-results-between-adaptive-time-step-and-constant-dt.9186/).

Reference:

De Morais, M.V.B., and Guerrero, V.V.U. (2018). Analysis of Computational Performance and Adaptive Time Step for Numerical Weather Prediction Models. Int. J. Eng. Math. Model, 2018, 1–8.

(3) Section 2.3. - Could the authors provide briefly more details about the interpolation method used in the RegRAIN package? That could ease the reading and understanding of that part of the text.

**Reply:** RegRAIN is a regionalized rain interpolator model based on the Regionalisierte Niederschlage (REGNIE) method (Rauthe et al., 2013). REGNIE is a combination between multiple linear regression (MLE) considering orographical conditions (e.g., latitude and longitude, slope, and elevation) and inverse distance weighting. To do that, a digital elevation model (DEM) is used together with monthly precipitation time series from stations. These details will be included in the new version of the manuscript In Section 2.3 according to the reviewer's suggestion.

Reference:

Rauthe, M., Steiner, H., U., Riediger, A., Mazurkiewicz, A., & Gratzki, A. (2013). A Central European precipitacion climatology–Part I: Generation and validation of a high-resolution gridded daily data set (HYRAS). Meteorologische Zeitschrift, 22(3), 235-256. https://doi.org/10.1127/0941-2948/2013/0436

(4) Sections 2.4.4, 2.4.5, and 2.4.7 - I am not sure if the units of the variables related to precipitation are correctly stated in these two sections. For example, Precipitation is defined between brackets as kg/m2, but then in the explanation, it is defined as mm/hour. This also happens in BIO12, BIO13, BIO19, Wet-hour Intensity and Maximum amount of precipitation in the wettest month. Could the authors check these mismatches? Is the definition of the Simple Daily Intensity Index correct? It is defined as the mean annual pr when pr >1mm, but I would guess that it should be related to daily values instead of annual values. Please check that.

**Reply:** Thank you. This was a mistake that will be corrected in the new version of the manuscript. Precipitation variables as reported in … Protocol is given in kg m$^{-2}$. SDII is the sum of daily precipitation for a given period of time (here we used a year) divided by wet days (days with pr > 1 mm). We agree with the reviewer and the definition will be changed to:

*"SDII calculates the mean pr for wet days (pr > 1 mm) …"*

(5) Section 3.1 - The manuscript would benefit from a short explanation of the pseudo-PDFs. More concretely, why and how they are calculated. Lines 317-320: I missed a comment about the shape of the probability distribution function shown by ERA5-Land compared to the other datasets. I would say that it is also different around 20ºC.
**Reply:** As suggested by the reviewer, and also by the first referee, more detail will be given on how the pseudo-PDFs were made.

(6) Section 3.2 -Line 353: Can the authors elaborate more on what they mean by "amount of energy"? That sentence is referred to BIO1, and that is the annual mean temperature, so it is not an energy
**Reply:** This sentence is according to the definition in other works such as Noce et al. (2020), where BIO1 is defined as "the total amount of energy inputs for the ecosystems in a year" so with energy we wanted to say energy available for ecosystems. We will rewrite this part in the new version of the manuscript to clarify this point.

(7) Figure 4: There is a mismatch between the labels in the Figure and the Caption. UGR-SNGrid is labelled as GFAT-Grid in the figure. The same happens in Figure S7 in the supplementary.
**Reply:** Thank you for pointing out this error, which will be corrected in the new version of the manuscript.

**Minor comments:**

- References should be listed in chronological order throughout the manuscript (e.g., lines 46,48-49, 68-69, etc).
  **Reply:** According to the in-text citations rules (https://www.earth-system-science-data.net/submission.html#references), the order can be based on relevance, as well as chronological or alphabetical listing, depending on the author's preference. Here an alphabetical order is chosen.

- Line 76: I would start a new paragraph to explain the structure of the publication.
  **Reply:** as suggested by the reviewer, this change will be made in the new version of the manuscript.

- Line 110: initial soil moisture conditions or soil moisture initial conditions.
  **Reply:** In this case we intend to say, initial soil moisture conditions. This will be changed in the new version of the manuscript.

- Line 156: 200 m instead of two hundred.
  **Reply:** This aspect will be changed according to the referee's suggestion in the new version of the manuscript.

- Table 2: I printed the PDF and it seems that there is a problem with the formatting of the text related to the coverage of CHIRPS. It appears as bold.
  **Reply:** We have checked this text and it is not in bold.

- Table 2: Define ta in the caption, as it is not explained in the text yet. In this version, only pr, tasmax and tasmin are explained in the caption.
  **Reply:** ta (mean temperature) will be defined in the new version of the manuscript.
- Line 200: kg*kg-1
  **Reply:** This will be fixed according to the referee's suggestion in the new version of the manuscript.
- Lines 238-239: "..., and then minimum values for each year are taken AS the minimum..."
  **Reply:** This will be fixed according to the referee's suggestion in the new version of the manuscript.
- Line 270: I suggest removing the * as the note is in the following line.
  **Reply:** The asterisk was added because it is a slightly different definition from the conventional one. However, as suggested by the reviewer, the asterisk will be removed.
- Page 13: Table 3 is defined as Table 1 again
  **Reply:** This will be fixed according to the referee's suggestion in the new version of the manuscript.
- Line 334: The highest precipitation is found in December, right?
  **Reply:** right, this will be fixed according to the referee's suggestion in the new version of the manuscript.
- Line 397 – Caption Fig. 4: It states temperature, but it should be precipitation recorded by the SAIH stations.
  **Reply:** right, this will be fixed according to the referee's suggestion in the new version of the manuscript.
- Line 418: BIO5, BIO8 and BIO9, according to the subfigures mentioned in that line.
  **Reply:** right, this will be fixed according to the referee's suggestion in the new version of the manuscript.
- Lines 337-338: Something is missing in the sentence. Otherwise, I suggest rewriting it.
  **Reply:** The sentence will be rewritten according to the reviewer's suggestion.
- Line 455: With values between 30 and 140 mm.
  **Reply:** This will be changed according to the referee's suggestion in the new version of the manuscript.

---

## Author Comment (AC2)

**Reply to Anonymous Referee #1**

This work provides a 1km-resolution climate dataset from January 1991 to December 2022 for the Sierra Nevada mountain range in Spain. This high-resolution dataset will help to study the impacts of climate change on the botany, ecology and other aspects of mountainous regions.

The dataset was generated using the Weather Research and Forecasting (WRF) model with lateral boundary conditions given by the ERA5 reanalysis. The configuration of the WRF model is clearly described. The quality of the dataset is carefully evaluated. The climate variables of the dataset are well introduced.

**Reply:** Authors thank the reviewer for reading the manuscript and taking the time to review it, asking critical questions, which will help us to improve the quality of the manuscript. This comment will be therefore included in the acknowledgements section. We have responded to the referee's request point by point indicating the actions to be taken in the final revision of the manuscript. All our replies are included in blue.

**General comments:**

(1) Do the authors think that assimilating station data into WRF simulations will improve data quality? There could be some discussion at the end of the paper.

**Reply:** We think that the use of data assimilation would be of interest. However, for such assimilation, observational information for the region is required, and in this respect, note that Sierra Nevada is a region of complex topography where observational data are very scarce, already in the case of temperature and precipitation, and even more so for other variables, and if they exist, they are of short temporal length and low quality. For this reason, we believe that at present we cannot generate quality climate information using assimilation techniques. Everything indicated here will be included as a discussion at the end of the article as suggested by the reviewer.

(2) The authors clearly described how the WRF model is configured and listed relevant references. However, as a non-expert in this model, I would have appreciated a bit more information on the justification of the configuration chosen by the authors and how well the WRF model simulates mountain climates in general.

**Reply:** An important part of the configuration was based on the balance between obtaining good quality results and the computational cost, since CPM simulations require a large amount of resources, especially when using a model such as WRF. For example, the configuration of the domains was based on the experience of the research group. Configurations based on a single domain require simulating a much larger domain as the change in resolution from ERA5 to 1 km is very pronounced. However, the use of two domains with a 1:5 grid ratio seems to be adequate and has already been employed by other researchers such as Messmer et al. (2021). Moreover, we used a one-way nesting approach to complete CMP simulations because it has a lower computational cost than two-way nesting and the improvement of the latter is not very appreciable according to several studies such as Prein et al. (2015) or Messmer et al. (2021). Concerning vertical levels, we selected 46 vertical levels since our experience indicates that a greater number of levels does not generate large differences in the results, but they do involve a great computational effort. On the other hand, we use

hybrid coordinates because they reduce numerical errors associated with the influence of topography. The model was fed with initial and boundary conditions every six hours as it has proven to be a suitable frequency, also for CPM simulations. In this respect there are authors who indicate that higher frequencies could be more suitable but when we talk about climate simulations the cost/benefit seems to be unclear. All these details will be added in the new version of the manuscript considering the reviewer's suggestion.

Similarly, and taking into account the suggestions of referee 2, we will include additional information from the study used as a reference to selecting the physics schemes in order to make clearer the aspects related to the configuration of the model.

Moreover, we will add more information about the ability of a regional climate model (RCM) such as WRF for simulating mountain climates in general.

References

Messmer, M., González-Rojí, S. J., Raible, C. C., and Stocker, T. F.: Sensitivity of precipitation and temperature over the Mount Kenya area to physics parameterization options in a high-resolution model simulation performed with WRFV3.8.1, Geosci. Model Dev., 14, 2691–2711, https://doi.org/10.5194/gmd-14-2691-2021, 2021.

Prein, A. F., Langhans, W., Fosser, G., Ferrone, A., Ban, N., Goergen, K., Keller, M., Tölle, M., Gutjahr, O., Feser, F., Brisson, E., Kollet, S., Schmidli, J., Van Lipzig, N. P. M., and Leung, R.: A review on regional convection-permitting climate modeling: Demonstrations, prospects, and challenges, Reviews of Geophysics, 53, 323–361, https://doi.org/10.1002/2014RG000475, 2015.

(3) There seems to be some data (daily primary climate variables) missing at
   o Longitude: -3.296478271484375, Latitude: 37.09039306640625
   o Longitude: -3.352783203125, Latitude: 37.099708557128906
   o Longitude: -3.465118408203125, Latitude: 37.16318893432617
   o Longitude: -3.476409912109375, Latitude: 37.16325378417969

HighResClimNevada.GFAT-UGR.ECMWF-ERA5.Evaluation.WRF433.day.tasmax.1991010100-2022123100.nc (day 1)

**Reply:** Thank you for your appreciation. There are some points with missing data because this database was created using the land-sea mask from WRF. In those points where WRF detect "sea" (a lake in this case), the temporal series was masked. In this regard, we will add an additional comment in the data description to clarify that this data is only for land.

**Specific comments:**

- Line 23-24: "For precipitation, variable, more uncertain and difficult to characterize, HighResClimNevada exhibits a higher amount of precipitation when compared to station-based, coarse satellite-based, and reanalysis-based products." This relates to the second general comment. How sensitive are the results to the model configuration? Is the chosen configuration optimal?

  **Reply:** WRF is sensitive to the model configuration and the simulation was completed using a configuration particularly designed in convection-permitting mode, considering aspects related to the domain, the number of vertical levels and the parameterization settings, among others. The latter plays a very relevant role in the results, so it is one of the most tested aspects in our study. For this reason, a sensitivity study was obtained

from these tests, which was recently published in Atmospheric Research (Please see Solano-Farías et al., 2024), from which the best set of parameterizations for the region was selected. As indicated in the general comments, more details will be included in the new version of the manuscript.

Reference:

Solano-Farias, F., García-Valdecasas Ojeda, M., Donaire-Montaño, D., Rosa-Cánovas, J. J., Castro-Díez, Y., Esteban-Parra, M. J., and Gámiz-Fortis, S. R.: Assessment of physical schemes for WRF model in convection-permitting mode over southern Iberian Peninsula, Atmospheric Research, 299, 107175, https://doi.org/10.1016/j.atmosres.2023.107175, 2024.

- Line 176-177: Why were temperature observations from only two stations used?

  **Reply:** The stations with temperature data available are scarcer than for precipitation. At the same time, when we did the quality check, we only obtained two stations with 19 years of data with at least 85% of records (Line 175-176). In this regard, additional details will be included in Table 2.

- Table 2: Is there information on the number of stations in the second column?

  **Reply:** Yes, the second column in the case of punctual in-situ stations between parentheses indicates the maximum number of stations used as indicated in its header. This was done considering the number of stations in general, since those stations providing temperature are contained in precipitation. In this sense, the value for SAIH-S will be changed to also indicate the stations contemplated only for temperature.

- Line 314-315: In addition to the citation, is it possible to describe briefly how to obtain the pseudo-PDFs?

  **Reply:** To obtain the pseudo-PDFs, the precipitation was grouped by events with bin of 1 mm, all the precipitation accumulated in the grid points within the National and Natural parks was computed, dividing it by the number of points and the number of days. In this regard, we will describe briefly how the pseudo-PFDs were obtained in the new version of the manuscript.

- Line 480: Is it possible to provide a link to the data download page?

  **Reply:** We are not quite sure which link the reviewer is referring to. The only link that seems to be missing could be in reference to the stations which will be included in the new version of the manuscript.

- Figure 6: Black is used to define two things.

  **Reply:** Thank you for pointing this out. Actually, black color is only used to indicate the HighResClimNevada position in each square, but there is a mistake in the figure caption and the years without information are shown in Gray. We will fix this issue in the next version of the manuscript.

**Technical corrections:**

- Line 38 and 40: Missing space at "(Parmesan, 2006). For" and "(Beniston, 2003).This"

  **Reply:** Thank you, this will be corrected.

- Line 76-79: Inconsistent use of "." and ";"

  **Reply:** Thank you, this will be corrected.

- Figure 4: One dataset name in the caption and plot title does not match (GFAT-grid -> UGR-SNGrid)

  **Reply:** Thank you for your appreciation, a new Figure 4 will be made with the correct name.

---

## Author Response (AR1)

**Reply to Anonymous Referee #1**

This work provides a 1km-resolution climate dataset from January 1991 to December 2022 for the Sierra Nevada Mountain range in Spain. This high-resolution dataset will help to study the impacts of climate change on the botany, ecology and other aspects of mountainous regions.

The dataset was generated using the Weather Research and Forecasting (WRF) model with lateral boundary conditions given by the ERA5 reanalysis. The configuration of the WRF model is clearly described. The quality of the dataset is carefully evaluated. The climate variables of the dataset are well introduced.

Authors thank the reviewer for reading the manuscript and taking the time to review it, asking critical questions, which will help us to improve the quality of the manuscript. This comment will be therefore included in the acknowledgements section. We have responded to the referee's request point by point indicating the actions to be taken in the final revision of the manuscript. All our replies are included in blue.

**General comments:**

(1) Do the authors think that assimilating station data into WRF simulations will improve data quality? There could be some discussion at the end of the paper.

**Reply:** We think that the use of data assimilation would be of interest. However, for such assimilation, observational information for the region is required, and in this respect, note that Sierra Nevada is a region of complex topography where observational data are very scarce, already in the case of temperature and precipitation, and even more so for other variables, and if they exist, they are of short temporal length and low quality. For this reason, we believe that at present we cannot generate quality climate information using assimilation techniques. Everything indicated here has been included as a discussion at the end of the article as suggested by the reviewer.

"*Therefore, HighResClimNevada has been developed with the purpose of filling the gap of lack of long-term climate information regular in space and time using climate modeling. This database has been generated based only on climate model outputs and therefore it does not consider information from observations of ground-based stations or satellites via assimilation in order to avoid new sources of uncertainties. Note, observations in this region are usually short and contain errors due to instrumental inaccuracies or poorly calibrated equipment since SN is an area of difficult access. Furthermore, satellite information frequently has substantial uncertainties and short records; hence, assimilation may contribute additional uncertainty into climate data*"

Please see lines from 500 to 506 in the revised manuscript.

(2) The authors clearly described how the WRF model is configured and listed relevant references. However, as a non-expert in this model, I would have appreciated a bit more information on the justification of the configuration chosen by the authors and how well the WRF model simulates mountain climates in general.

**Reply:** An important part of the configuration was based on the balance between obtaining good quality results and the computational cost, since convection-permitting simulations require a large amount of resources, especially when using a model such as WRF. For example, the configuration of the domains was based on the experience of the research group. Configurations based on a single domain require simulating a much larger domain as the change in resolution from ERA5 to 1 km is very pronounced. However, the use of two domains with a 1:5 grid ratio seems to be adequate and has already been employed by other researchers such Gonzalez-Rojí et al. (2022). Moreover, we used a one-way nesting approach to complete convection-permitting simulations because it has a lower computational cost than two-way nesting and the improvement of the latter is not very appreciable according to several studies such as Messmer et al. (2021).

Concerning vertical levels, we selected 46 vertical levels since our experience indicates that a greater number of levels does not generate large differences in the results, but they do involve a great computational effort. On the other hand, we use hybrid coordinates because they reduce numerical errors associated with the influence of topography.

The model was fed with initial and boundary conditions every six hours as it has proven to be a suitable frequency, also for CPM simulations. In this respect there are authors who indicate that higher frequencies could be more suitable but when we talk about climate simulations the cost/benefit seems to be unclear. All these details will be added in the new version of the manuscript considering the referee's suggestion. Moreover, Gonzalez Rojí argued the 6 hours is a good update frequency when we want to achieve certain degree of freedom.

Similarly, and considering the suggestions of referee 2, we will include additional information from the study used as a reference to selecting the physics schemes to make clearer the aspects related to the configuration of the model.

We have also added a sentence indicating the behavior of this model in regions of complex topography.

Please see Section 2.2 (Climate model)

References

Messmer, M., González-Rojí, S. J., Raible, C. C., and Stocker, T. F.: Sensitivity of precipitation and temperature over the Mount Kenya area to physics parameterization options in a high-resolution model simulation performed with WRFV3.8.1, Geosci. Model Dev., 14, 2691–2711, https://doi.org/10.5194/gmd-14-2691-2021, 2021.

González-Rojí, S. J., Messmer, M., Raible, C. C., and Stocker, T. F.: Sensitivity of precipitation in the highlands and lowlands of Peru to physics parameterization

options in WRFV3.8.1, Geosci. Model Dev., 15, 2859–2879, https://doi.org/10.5194/gmd-15-2859-2022, 2022.

(3) There seems to be some data (daily primary climate variables) missing at
- Longitude: -3.296478271484375, Latitude: 37.09039306640625
- Longitude: -3.352783203125, Latitude: 37.099708557128906
- Longitude: -3.465118408203125, Latitude: 37.16318893432617
- Longitude: -3.476409912109375, Latitude: 37.16325378417969

HighResClimNevada.GFAT-UGR.ECMWF-ERA5.Evaluation.WRF433.day.tasmax.19910101 00-2022123100.nc (day 1)

**Reply:** Thank you for your comment. There are some points with missing data because this database was created using the land-sea mask from WRF. In those points where WRF detect "sea" (a lake in this case), temporal series have been masked. To make it clear, we have included an extra comment in the data description, as follows:

*"All files also provide a 2D mesh with the altitude (z) at sea level expressed in meters above the mean sea level and includes variables for over land."*

Please see lines 192-193 in the revised manuscript.

**Specific comments:**

- Line 23-24: "For precipitation, variable, more uncertain and difficult to characterize, HighResClimNevada exhibits a higher amount of precipitation when compared to station-based, coarse satellite-based, and reanalysis-based products." This relates to the second general comment. How sensitive are the results to the model configuration? Is the chosen configuration optimal?

    **Reply:** Right, precipitation is more uncertain and difficult to characterize for the model, but also for reference datasets, satellite products, and reanalysis. Note that the development of observational gridded datasets such as ROCIO_IBEB is a challenge due to several factors such as the lack of observational data, which, if any, has a limited quality, the spatial variability and the sharp orographic gradient make difficult to accurately represent the spatiotemporal distribution of precipitation. The Sierra Nevada topography makes installing and maintaining weather stations difficult. Therefore, higher uncertainties are expected in the development of gridded products, and we need to use different sources of precipitation estimates for the evaluation of this variable being the spatial resolution a factor with a major role.

    WRF is sensitive to the model configuration and the simulation was completed using a configuration particularly designed in convection-permitting mode, considering aspects related to the domain, the number of vertical levels and the parameterization settings, among others. The latter plays a very relevant role in the results, so it is one of the most tested aspects in our study. For this reason, a sensitivity study was obtained from these tests, which was recently published in Atmospheric Research (Please see Solano-Farías et al., 2024), from which the best set of parameterizations for the region was selected. As indicated in the general comments, more details will be included in the new version of the manuscript, which are as follows.

*"This configuration was achieved by performing a sensitivity study resulting from combining different microphysics and convection schemes in the parent domain (d01), i.e., convection was switched off in the inner domain (d02) for all experiments. As a result, 12 WRF simulations of 1 year length were completed and compared with different reference datasets in terms of precipitation and maximum and minimum temperatures"*

Please see lines 120-124 in the revised manuscript.

Reference:

Solano-Farias, F., García-Valdecasas Ojeda, M., Donaire-Montaño, D., Rosa-Cánovas, J. J., Castro-Díez, Y., Esteban-Parra, M. J., and Gámiz-Fortis, S. R.: Assessment of physical schemes for WRF model in convection-permitting mode over southern Iberian Peninsula, Atmospheric Research, 299, 107175, https://doi.org/10.1016/j.atmosres.2023.107175, 2024.

- Line 176-177: Why were temperature observations from only two stations used?

  **Reply:** SAIH stations with temperature data available were scarcer than for precipitation and when we did the quality check, we only obtained two stations with 19 years of data with at least 85% of records (Line 179 in the revised manuscript). In this regard, additional details will be included in Table 2 to clearly show the number of stations for temperature evaluation.

- Table 2: Is there information on the number of stations in the second column?

  **Reply:** Yes, the second column in Table 2 shows the spatial coverage together with the resolution for gridded products and number of stations for in-situ data, the latter in brackets. This was done considering the number of stations in general, since those stations providing temperature are contained in precipitation. In this sense, the value for SAIH-S will be changed to also indicate the stations contemplated only for temperature. Also, the Table 2 caption has been changed to clarify this point.

  Please see the Table 2 caption in the revised manuscript.

- Line 314-315: In addition to the citation, is it possible to describe briefly how to obtain the pseudo-PDFs?

  **Reply:** According to the referee's suggestion, we have included a brief description of how the pseudo-PDF has been calculated. In this sense, pseudo-PDFs were calculated by grouping events using a bin of 2 mm. All the precipitation fallen in grid points within the National and Natural Park borders was accumulated, and then, the obtained value was divided by the number of grid points and the number of days. Please see lines from 308 to 312 in the revised manuscript:

*"That is, considering all grid points within the National and Natural Park borders, pseudo-PDFs for each dataset were obtained by grouping events of daily precipitation (pr > 0.1 mm) into 2 mm bins. The number of events multiplied by the mean intensity for each bin was then determined by dividing the total precipitation amounts (expressed in mm) for each bin by the number of grid points and days. Pseudo-PDFs were selected instead of the traditional PDFs in order to avoid masking light precipitation and, more importantly, heavy precipitation events."*

- Line 480: Is it possible to provide a link to the data download page?

  **Reply:** We are not quite sure which link the reviewer is referring to. The only link that seems to be missing could be in reference to the stations which will be included in the revised manuscript.

- Figure 6: Black is used to define two things.

  **Reply:** Thank you for pointing this out. Actually, black color is only used to indicate the HighResClimNevada position in each square, but there is a mistake in the figure caption and the years without information are shown in Grey. Therefore, the figure caption has been fixed as well as the figure description (line 428 in the revised manuscript) according to the referee's suggestion, changing black to grey. Please see the caption of Fig. 6 in the revised manuscript.

**Technical corrections:**

- Line 38 and 40: Missing space at "(Parmesan, 2006). For" and "(Beniston, 2003). This"

  **Reply:** Thank you, the typo has been corrected. Please see lines 39 y 43 in the revised manuscript.

- Line 76-79: Inconsistent use of "." and ";"

  **Reply:** Thank you, the typo has been corrected. Please see lines 76-79 in the revised manuscript.

  Figure 4: One dataset name in the caption and plot title does not match (GFAT-grid -> UGR-SNGrid)

  **Reply:** Thank you for your appreciation, a new Figure 4 has been made with the correct name.

**Reply to Anonymous Referee #2**

In this study, the authors present a new high-resolution dataset over Sierra Nevada in the Iberian Peninsula for the period 1991-2022. The dataset was created with the WRF model, and ERA5 reanalysis data provided the initial and boundary conditions. To test the accuracy of the model, many indices were compared against alternative datasets based on observations and satellite data. The reliable performance of the dataset is highlighted throughout the manuscript. This dataset is valuable for the climate research community working on high mountain environments, but also for other areas such as biology or ecology.

The manuscript follows a logical structure and fits into the scope of Earth System Science Data. However, the authors need to address some comments before it is ready for publication.

Authors thank the reviewer for reading the manuscript and taking the time to review it, asking critical questions, which will help us to improve the quality of the manuscript. This comment will be therefore included in the acknowledgements section. We have responded to the referee's request point by point indicating the actions to be taken in the final revision of the manuscript. All our replies are included in blue.

**Major comments:**

(1) Not related to any section: - After reading the paper, it is unclear if the variables are provided in the original Lambert grid from WRF, or if they have been interpolated to a regular lon lat grid. That should be clearly stated in the text. That is my concern since wind speed is calculated using equation 5, and both components depend on the grid in which they are provided. For a regular lon lat grid, the winds should be rotated from WRF's Lambert grid.

**Reply:** we are sorry for the misunderstanding. The variables are provided according to the original Lambert grid provided by the model. In this regard we have added new details to the revised manuscript saying that the data are in their original mesh and therefore it has not been necessary to rotate the wind. The text included in the revised manuscript is as follows:

*"These data are provided in the original Lambert grids for a period from January 1991 to December 2022."*

Please see lines 191-192 in the revised manuscript.

(2) Section 2.2. - The authors could briefly explain the model sensitivity analysis that led to the selected configuration in WRF. Only the reference is provided in the current version, but the manuscript can improve if more details are provided. For example: Was the analysis based on temperature? Precipitation? Other variables? - Line 111 mentions the trade-off between suitability and computational resources, but Table 1 provides specific time steps for each domain. WRF can run using an adaptive time step to reduce the simulations' running time. Was the simulation created using that option? If not, is there any reason why the authors did not use it? - I would consider a 5 km spatial resolution already a convective scale. Thus, I do not see a reason to use the convection parameterization in the first domain of the model. Did the authors test that alternative?

**Reply:** This sensitivity study aimed to analyze the WRF model performance over Andalusia (southern Spain) using 12 1-year simulations resulting from combining different microphysics and cumulus options in the parent domain (d01) (i.e., the cumulus scheme was only switched off in the inner domain). These experiments were evaluated in terms of precipitation and maximum and minimum temperatures using different data as reference. In this study, moreover, by answering one of the referee's questions, among the different configurations, one with convection switched off in the parent domain (d01) was also tested, but although this was one of the options that obtained better results, in general, the use of Grell Freitas convection in d01 seemed to yield more adequate results in all the analyzed variables. Regarding this study, as suggested by the reviewer, we will add more details about the sensitivity study performed in the new version of the manuscript.

More details about this study have been added to the manuscript.

"For the selection of physical schemes, we applied the configuration suggested by Solano-Farías et al. (2024). This configuration was achieved by performing a sensitivity study resulting from combining different microphysics and convection schemes in the parent domain (d01), i.e., convection was switched off in the inner domain (d02) for all experiments. As a result, 12 WRF simulations of 1 year length were completed and compared with different reference datasets in terms of precipitation and maximum and minimum temperatures."

Please see lines 120-124 in the revised manuscript.

On the other hand, a fixed time-step was used following the recommendations of the WRF model developers (6*spatial resolution). This type of timestep is widely used in WRF in convection-permitting simulations (e.g. Zhang et al., 2025), showing adequate results in the IP. However, it could be an aspect to investigate for future studies, thank you for this suggestion. In any case, this type of time step should be used with caution and as indicated in some studies (e.g., De Morais and Guerrero, 2018) appears to be less advisable than the use of fixed values. In addition, it seems that in long simulations there can be marked differences between the results of using one or the other type of time step, an aspect not solved at least until version 4.2 (please see the

https://forum.mmm.ucar.edu/threads/different-results-between-adaptive-time-step-and-constant-dt.9186/).

Reference:

De Morais, M.V.B., and Guerrero, V.V.U. (2018). Analysis of Computational Performance and Adaptive Time Step for Numerical Weather Prediction Models. Int. J. Eng. Math. Model, 2018, 1–8.

Zhang, Y., Deng, C., Xu, W. et al. (2025). Long-term variability of extreme precipitation with WRF model at a complex terrain River Basin. Sci Rep 15, 156.

(3) Section 2.3. - Could the authors provide briefly more details about the interpolation method used in the RegRAIN package? That could ease the reading and understanding of that part of the text.

**Reply:** RegRAIN is a regionalized rain interpolator model based on the Regionalisierte Niederschlage (REGNIE) method (Rauthe et al., 2013). REGNIE is a combination between multiple linear regression (MLE) considering orographical conditions (e.g., latitude and longitude, slope, and elevation) and inverse distance weighting. To do that, a digital elevation model (DEM) with 200 m spatial resolution, which is available at https://centrodedescargas.cnig.es/CentroDescargas/modelo-digital-terreno-mdt200-primera-cobertura, is used together with monthly precipitation time series from stations. These details have been included in the revised manuscript according to the referee's suggestion.

*"Based on the Regionalisierte Niederschlage (REGNIE) method (Rauthe et al., 2013), RegRAIN combines multiple linear regression considering orographical factors such as location, slope, elevation, and inverse distance weighting. The orographical factors were obtained from a digital elevation model and monthly regressions are calculated using precipitation time series from stations. More details about this methodology can be found in in Romero-Jiménez et al. (2023)."*

Please see lines 160-163 (Section 2.3) in the revised manuscript.

Reference:

Rauthe, M., Steiner, H., U., Riediger, A., Mazurkiewicz, A., & Gratzki, A. (2013). A Central European precipitacion climatology–Part I: Generation and validation of a high-resolution gridded daily data set (HYRAS). Meteorologische Zeitschrift, 22(3), 235-2561

(4) Sections 2.4.4, 2.4.5, and 2.4.7 - I am not sure if the units of the variables related to precipitation are correctly stated in these two sections. For example, Precipitation is defined between brackets as kg/m2, but then in the explanation, it is defined as mm/hour. This also happens in BIO12, BIO13, BIO19, Wet-hour Intensity and Maximum amount of precipitation in the wettest month. Could the authors check these mismatches? Is the definition of the Simple Daily Intensity Index correct? It is defined as the mean annual pr when pr > 1mm, but I would guess that it should be related to daily values instead of annual values. Please check that.

**Reply:** Thank you. This was a mistake that was corrected as is shown in the revised manuscript. Accumulated precipitation is given in kg m$^{-2}$ at both hourly and daily scale and in the case of BIO12, BIO13 And BIO19. On the other hand, SDII is the sum of daily precipitation for a given period of time (here we used a year) divided by wet days (days with pr > 1 mm). We agree with the reviewer and the SDII definition has been changed to:

*"SDII calculates the mean daily pr on wet days (pr >1 mm) for a year and rx5day is the highest total quantity of pr falling on 5 consecutive days"*

Please see lines 281-282 in the revised manuscript.

(5) Section 3.1 - The manuscript would benefit from a short explanation of the pseudo-PDFs. More concretely, why and how they are calculated. Lines 317-320: I missed a comment about the shape of the probability distribution function shown by ERA5-Land compared to the other datasets. I would say that it is also different around 20ºC.

**Reply:** We used Pseudo-PDFs instead of traditional PDFs to avoid masking of light precipitation, and in particular, heavy precipitation thus providing an easy and direct interpretation of the total precipitation (Argüeso et al., 2012). As suggested by the reviewer, and also by the first referee, more detail has been included in this regard:

*"That is, considering all grid points within the National and Natural Park borders, pseudo-PDFs for each dataset were obtained by grouping events of daily precipitation (pr > 0.1 mm) into 2 mm bins. The number of events multiplied by the mean intensity for each bin was then determined by dividing the total precipitation amounts (expressed in mm) for each bin by the number of grid points and days. Pseudo-PDFs were selected instead of the traditional PDFs in order to avoid masking light precipitation and, more importantly, heavy precipitation events."*

Please see lines 307-311 in the revised manuscript.

Reference:
Argüeso, D., Hidalgo-Muñoz, J. M., Gámiz-Fortis, S. R., Esteban-Parra, M. J., and Castro-Díez, Y.: Evaluation of WRF Mean and Extreme Precipitation over Spain: Present Climate (1970–99), Journal of Climate, 25, 4883–4897, https://doi.org/10.1175/JCLI-D-11-00276.1, 2012.

(6) Section 3.2 -Line 353: Can the authors elaborate more on what they mean by "amount of energy"? That sentence is referred to BIO1, and that is the annual mean temperature, so it is not an energy

**Reply:** This sentence is according to the definition in other works such as Noce et al. (2020), where BIO1 is defined as "the total amount of energy inputs for the ecosystems in a year" so with energy we wanted to say energy available.

(7) Figure 4: There is a mismatch between the labels in the Figure and the Caption. UGR-SNGrid is labelled as GFAT-Grid in the figure. The same happens in Figure S7 in the supplementary.

**Reply:** Thank you for pointing out this error, Figure 4 and S7 have been fixed according to the referee's comment, and now SNGrid is labelled correctly. The mistake has been also corrected in Zenodo (https://doi.org/10.5281/zenodo.14883471) where a revised version of the supplementary material has been included.

**Minor comments:**

- References should be listed in chronological order throughout the manuscript (e.g., lines 46,48-49, 68-69, etc).
  **Reply:** According to the in-text citations rules (https://www.earth-system-science-data.net/submission.html#references), the order can be based on relevance, as well as chronological or alphabetical listing, depending on the author's preference. Here an alphabetical order is chosen.

- Line 76: I would start a new paragraph to explain the structure of the publication.
  **Reply:** as suggested by the referee, the structure of the publication has been detailed in a new paragraph. Please see lines from 76 to 79 in the revised manuscript.

- Line 110: initial soil moisture conditions or soil moisture initial conditions.
  **Reply:** In this case we intend to say, initial soil moisture conditions. Thus, the main text of the manuscript is now as follows:

  *"However, soil variables need longer periods to reach such an equilibrium, and it depends on the initial soil moisture conditions and soil depth, among others (Khodayar et al., 2015)."*

  This has been changed in the revised manuscript (line 117).

- Line 156: 200 m instead of two hundred.
  **Reply:** This aspect will be changed according to the referee's suggestion in the new version of the manuscript. Please see line 157 in the revised version of the manuscript.

- Table 2: I printed the PDF and it seems that there is a problem with the formatting of the text related to the coverage of CHIRPS. It appears as bold.
  **Reply:** Thank you for pointing this out. We have checked the coverage of CHIRPS, and it is not bold.

- Table 2: Define ta in the caption, as it is not explained in the text yet. In this version, only pr, tasmax and tasmin are explained in the caption.

**Reply:** ta acronym (hourly mean temperature) has been defined in the table 2 caption in the revised manuscript.

- Line 200: kg*kg-1
  **Reply:** This has been fixed according to the referee's suggestion in the revised manuscript (line 203).

- Lines 238-239: "..., and then minimum values for each year are taken AS the minimum..."
  **Reply:** This has been fixed according to the referee's suggestion in the new version of the manuscript (line 243).

- Line 270: I suggest removing the * as the note is in the following line.
  **Reply:** The asterisk was added because it is a slightly different definition from the conventional one. However, as suggested by the referee, the asterisk has been removed.

- Page 13: Table 3 is defined as Table 1 again
  **Reply:** This has been fixed according to the referee's suggestion in the revised manuscript (line 305).

- Line 334: The highest precipitation is found in December, right?
  **Reply:** right, this has been fixed according to the referee's suggestion in the revised manuscript (line 329).

- Line 397 – Caption Fig. 4: It states temperature, but it should be precipitation recorded by the SAIH stations.
  **Reply:** Caption 4 has been changed according to the referee's comments. Please see line 391 in the revised manuscript.

- Line 418: BIO5, BIO8 and BIO9, according to the subfigures mentioned in that line.
  **Reply:** right, this has been fixed according to the referee's suggestion in the revised manuscript (line 412).

- Lines 337-338: Something is missing in the sentence. Otherwise, I suggest rewriting it.
  **Reply:** The sentence has been rewritten according to the referee's suggestion, and now is as follows:

  **"***However, compared to reanalysis-based data like CERRA-LAND or ERA5-Land, HighResClimNevada seems to reveal a smaller pr amount throughout the summer.***"**

  Please see line 332-333 in the revised manuscript.

- Line 455: With values between 30 and 140 mm.
  **Reply:** This has been changed according to the referee's suggestion in the revised manuscript (line 449).

---

## Author Response (AR2)

**Reply to Editor**

I would like to thank the authors for their revisions, which I believe address the comments raised by the reviewers.

I have one request for the revision in L110 ("with the objective of acquiring certain degree of freedom (González-Rojí et al., 2022) ". Could you please rephrase or add an explanation to this.

**Reply:** The authors thank the editor for reading the manuscript and taking the time to review it, helping us to improve the quality of the manuscript.

In this study WRF was used to conduct regional climate simulations at very high spatial resolution. At this scale, the nesting methodology follows the same rules as at lower resolutions (Giorgi, 2019; Rummukainen, 2010). Some authors, such as Lucas-Picher et al. (2021), suggest that the update frequency of boundary conditions should be higher when we increase the spatial resolution. ERA5 provides hourly data so we could use data at that frequency to feed the model. However, since our objective is to perform climate simulations (not forecast), the use of boundary conditions at this frequency may be impractical due to the amount of storage and computational resources required. For this reason, in this case an update frequency of ERA5 every 6 hours was selected which seems acceptable for this study following the approach of Gonzalez-Rojí et al. (2022). In that case, the authors argued that they use data every six hours in order to allow the model to reach a certain degree of freedom by a lower update frequency of boundary conditions.

We have rewritten the paragraph to clarify the explanation as suggested by the editor, which we have marked in blue as was done in the previous revision. Please see L110-L114 in the new version of the revised manuscript.

"*WRF was updated every six hours using the fifth-generation European ReAnalysis (ERA5, Hersbach et al., 2020), the latest reanalysis product from the European Centre for Medium-Range Weather Forecasts, which has been shown to be effective for regional climate downscaling. Although ERA5 provides hourly data, we opted for a six-hour update frequency to allow the model a certain degree of free evolution (González-Rojí et al., 2022). Additionally, using hourly data for a 32-year climate simulation would be impractical due to the high storage and computational demands.*"

**References:**

Giorgi, F.: Thirty Years of Regional Climate Modeling: Where Are We and Where Are We Going next?, JGR Atmospheres, 124, 5696–5723, https://doi.org/10.1029/2018JD030094, 2019.

González-Rojí, S. J., Messmer, M., Raible, C. C., and Stocker, T. F.: Sensitivity of precipitation in the highlands and lowlands of Peru to physics parameterization options in WRFV3.8.1, Geosci. Model Dev., 15, 2859–2879, https://doi.org/10.5194/gmd-15-2859-2022, 2022.

Lucas-Picher, P., Argüeso, D., Brisson, E., Tramblay, Y., Berg, P., Lemonsu, A., Kotlarski, S., and Caillaud, C.: Convection-permitting modeling with regional climate models: Latest developments and next steps, WIREs Climate Change, 12, e731, https://doi.org/10.1002/wcc.731, 2021.

Rummukainen, M.: State-of-the-art with regional climate models, WIREs Climate Change, 1, 82–96, https://doi.org/10.1002/wcc.8, 2010.